# The Effect of Diet and Lifestyle on the Course of Diabetic Retinopathy—A Review of the Literature

**DOI:** 10.3390/nu14061252

**Published:** 2022-03-16

**Authors:** Anna Bryl, Małgorzata Mrugacz, Mariusz Falkowski, Katarzyna Zorena

**Affiliations:** 1Department of Ophthalmology and Eye Rehabilitation, Medical University of Bialystok, 15-089 Bialystok, Poland; malgorzata.mrugacz@umb.edu.pl; 2PhD Studies, Medical University of Bialystok, 15-089 Bialystok, Poland; mariusz.falkowski@adres.pl; 3Department of Immunobiology and Environmental Microbiology, Medical University of Gdansk, 80-211 Gdansk, Poland; katarzyna.zorena@gumed.edu.pl

**Keywords:** diabetic retinopathy, diet, physical activity

## Abstract

Diabetes is a major social problem. As shown by epidemiological studies, the world incidence of diabetes is increasing and so is the number of people suffering from its complications. Therefore, it is important to determine possible preventive tools. In the prevention of diabetic retinopathy, it is essential to control glycemia, lipid profile and blood pressure. This can be done not only by pharmacological treatment, but first of all by promoting a healthy lifestyle, changing dietary habits and increasing physical activity. In our work, we present a review of the literature to show that physical exercise and an adequate diet can significantly reduce the risk of diabetes and diabetic retinopathy.

## 1. Introduction

Diabetes is a major social issue. As recently reported by the International Diabetes Federation, the prevalence of diabetes (DM) worldwide was 9.3% (463 million) in adults between the ages of 20–79 years in 2019, with an anticipated rise to 10.9% (700.2 million) before 2045. This is a continuously growing burden for public health-care systems worldwide [1,2,3]. Although type 1 diabetes (T1DM) and type 2 diabetes (T2DM) vary in etiology, they are both associated with complications affecting the cardiovascular system, kidneys, eyes and nerves [4]. Cardiovascular disorders, heart failure, atherosclerosis and cerebrovascular incidents are mainly due to macrovascular lesions, while renal injury, retinal complications or neuropathy are caused by microvascular damage [4]. In both types, DM duration, glycemic control, arterial hypertension, creatinine levels and low-density lipoprotein (LDL) cholesterol were found to promote diabetic retinopathy (DR) [5]. The Vision Loss Expert Group reported in 2015 that globally diabetic retinopathy rose from the fifth to the sixth place of the most common causes of preventable vision defect [6], as compared to their earlier report from 2010 [7].

Numerous clinical studies confirm that physical activity and proper diet are major factors reducing the risk of diabetes and its ophthalmic complications [8]. Individuals who increased their physical activity and changed dietary habits reduced the development of this disease from 29 to 58% [9]. Here, we discuss possible diet modifications and the effect of lifestyle on the development and course of DR.

## 2. Diabetic Retinopathy

Clinically, diabetic retinopathy occurs as non-proliferative diabetic retinopathy (NPDR) and proliferative diabetic retinopathy (PDR). The former, being the initial stage of diabetic retinopathy, is manifested by severe retinal vascular permeability, formation of micro-aneurysms, exudation, hard exudate and hemorrhage. In such conditions, severe vision loss may develop due to neurosensory retinal atrophy and ischemia. In turn, PDR, being the advanced stage, presents with nerve-fiber layer infarcts, neovascular proliferation, vitreous tortuosity and hemorrhage, leading to the exacerbation of visual impairment caused by tractional retinal detachment and hemorrhage [10].

In 2010, the incidence of DR in the DM population worldwide was estimated to be around 34.6%, including 6.96% of PDR and 6.81% of diabetic macular edema (DME). Thus, a substantial number of DM patients had various grades of DR [11].

At the time of diagnosis, diabetic retinopathy is not observed in T1DM patients. This disorder develops later and after 20 years as many as 99% of diabetic patients have variously advanced symptoms. However, in T2DM patients DR can be already present when diabetes is diagnosed. Twenty years after the diagnosis of type 2 diabetes, approximately 60% of patients show symptoms of retinopathy [12]. When DR is in its early stage, the risk of vision loss is low and the only treatment that can be offered is to control the risk factors associated with the development and progression of this disorder. DR control aims primarily to regulate HbA1c and blood pressure, but also to reduce serum lipid levels and fight obesity [13,14,15]. This can be accomplished by conducting prophylactic examinations in order to diagnose the disease early and to tailor the therapy to individual patients, and above all by promoting a healthy lifestyle that can significantly affect the course of the disease.

## 3. Hemoglobin A1C (HbA1C < 7%) and Metabolic Memory

Metabolic memory plays a major role in the control of diabetic complications. This term refers to the phenomenon in which body tissues, including the retina, show reactions to poor or good control of glycemia years after improvement or exacerbation of glycemic control [16]. It has been shown that the tissues vulnerable to glucose-induced memory effects are the retinas, nerves and arteries [16]. A study conducted on animals revealed that after 2.5 years of poor glycemic control, at the time with no visible signs of retinopathy, diabetic dogs were subjected to good glycemic control for the subsequent 2.5 years. Within 5 years, these dogs developed severe diabetic retinopathy, similar to that observed in dogs with poor glycemic control throughout that period [17].

In humans, the metabolic memory of glycemia was first described in T1DM in the DCCT/EDIC study and in T2DM in a 10-year UKPDS. In the former study, metabolic memory for the management of intensive diabetes, reflecting a 2% lower HbA1c level for 6.9 years on average, was reported to prevent the development of diabetic retinopathy for more than 10 years after the study [18,19,20,21]. Further clinical research suggests that there is also a vascular metabolic memory referring to lipids and blood pressure [22].

Therefore, a major role should be assigned to early diagnosis and to the application of such therapeutic tools that could facilitate the maintenance of normal levels of these parameters.

The level of HbA1c is subject to memory. Adequate control of this parameter in time provides protection against DR progression in the case of any uncontrolled period [9]. Therefore, it is indispensable to assess this parameter early and frequently. A 1% decrease in HbA1c is related to a 35% reduction in the risk of DR development, 15–25% in the disease’s progression, 25% in visual acuity loss and 15% in the development of blindness [23]. In T1DM, in comparison to HbA1c accounting for 9%, HbA1c below 7% decreases the risk of DR development in 75% of cases and progression in 50%. Despite the significance of HbA1c reduction, the decrease should not be rapid since it may promote both hypoglycemia and DR progression [24]. A value below 6.5% is suggested in cases with a potential risk of nephropathy and DR. A value within the range of 7.1–8.5% can be acceptable in patients suffering from many coexisting diseases and if the optimum treatment fails to achieve a value less than 7% [25].

## 4. Blood Pressure (BP)

Hypertension has been found to enhance glycemia-induced oxidative stress, thus inducing more severe molecular lesions [8]. According to the pharmacological guidelines, antihypertensive treatment should be instituted in diabetic adults if their mean SBP is ≥130 mmHg or DBP ≥ 80 mmHg, and the aim of the treatment is to reach an SBP < 130 mmHg and DBP < 80 mmHg [26]. The meta-analysis of randomized controlled studies with diabetic patients revealed a statistically significant decrease in the risk of stroke, retinopathy and albuminuria when treatment for arterial hypertension was initiated at SBP < 140 mmHg and stopped when SBP < 130 mmHg was obtained [27]. The comparison of patients with BP 180/100 mmHg and those with BP 150/85 mmHg, revealed a 33% decrease in DR progression and in the need for laser application as well as a 50% reduction in loss of vision in the latter group [28].

Opposite to glycemia control, BP control has no memory effect. When BP control is decompensated, the risk of disease progression increases irrespective of the previous control [25].

## 5. Lipids

Its definition describes hyperlipidemia as low-density lipoprotein (LDL), total cholesterol, triglyceride or lipoprotein levels greater than the 90th percentile in comparison to the general population or an HDL level less than the 10th percentile when compared to the general population. Lipids typically include cholesterol levels, lipoproteins, chylomicrons, VLDL, LDL, apolipoproteins and HDL. The disorder has been found to correlate with some cardiometabolic disorders, including obesity, hypertension and coronary heart disease [29]. Although opinions of researchers on the correlation of hyperlipidemia and diabetic retinopathy are divided, in the majority of publications hyperlipidemia is considered a risk factor of diabetic retinopathy progression, including DME [30,31,32,33,34]. This supports the hypothesis that the inclusion of pharmacological treatment that normalizes lipid levels will have a beneficial effect on the development and course of changes in the retina. Statins have the potential to decrease total cholesterol and LDL-C, while fibrates may reduce TG. It has been observed that statins reduce the incidence of hard exudates and microaneurisms and decrease the risk of vision loss [35,36,37].

Fenofibrate also prevents the progression of diabetic retinopathy. It has been found to decrease its progression in T2DM and the necessity for laser therapy [38,39]. The combined treatment with fibrat and statins additionally diminished diabetic retinopathy by approximately 40% in comparison to patients taking statins only [39,40].

## 6. Obesity

Obesity is a growing problem of public health. It is associated with the development of many diseases [41] and is a major cause of mortality worldwide [42]. A significant correlation has been observed between obesity and the risk of diabetes [43]. The meta-analysis issued in 2018 revealed that obesity increased the incidence of DR in patients with T2DM. Obesity was not found to be related to PDR [42,44,45,46,47].

## 7. Stimulants

A correlation has been found between smoking and DR for T1DM but not for T2DM [48]. Occasional alcohol consumption was associated with a lower risk of DR than in non-drinkers [49]. Other researchers obtained similar results [50,51,52]. Beulens et al. revealed that moderate alcohol consumption was associated with decreased risk of microvascular complications in patients with T1DM [53]. The Beijing Eye Study suggested that alcohol intake was related to a reduced risk of DR in the general population [54]. Fenwick et al. showed that moderate drinking of white wine was correlated with a decreased risk of DR in patients with T2DM [55]. The beneficial effects induced by moderate alcohol consumption included increased levels of high-density lipoproteins, decreased platelet aggregation and a fall in the level of fibrinogen [56]. However, alcohol may have an effect on proinflammatory reaction and oxidative stress, which are significantly related to the risk of DR [57,58]. It should be remembered that alcohol addiction may lead to neglect of treatment, thus increasing the risk of DR development.

It has been shown that tea acts as a potent neuroprotector in the retina [59], preventing the formation of acellular capillary vessels and pericyte ghosts in diabetic rats [60]. Green tea can protect diabetic retinal neurons and regulate the subretinal environment by decreasing ROS generation due to the increased expression of glutamate transporter, restored intercellular connections and glutamine/glutamate circulation [61]. Moreover, a low dose of green tea is likely to improve antioxidant defenses, decrease inflammatory markers and prevent thickening of the basement membrane of the retina [62]. Black tea, which lowers blood sugar and thus inhibits pathological biochemical indices, is able to delay diabetic cataract development [63].

As found in a Chinese case–control study, involving sex- and age-matched controls suffering from diabetes but not diabetic retinopathy, regular consumption (every week for not less than 1 year) of Chinese green tea was associated with a reduced probability of retinopathy in women but not in men [64].

Coffee is one of the most consumed beverages worldwide. The main commercial coffee blends are Arabica (*Coffea arabica* L.) and Robusta (*Coffea canephora* Pierre ex Froehner) [65]. Its main component is caffeine, which belongs to antioxidants, and its long-term intake in large quantities reduces oxidative stress [66]. Other components of coffee beans include carbohydrates, proteins, fats, alkaloids, diterpenes, free amino acids, melanoides and minerals and both macro- and microelements [67]. Coffee contains microelements that show antioxidant effects, such as manganese, zinc, copper and iron [66]. It can also be a source of fluorine, chromium and cobalt [66].

A Norwegian cohort study revealed an approximately 35% reduced risk of T2D associated with high (in comparison to low) consumption of brewed coffee and other types of coffee [65]. Likewise, the results of a Finnish cohort study revealed that both the intake of boiled coffee and consumption of drip coffee were inversely correlated with T2D [67].

A number of coffee components have been found to show anti-inflammatory effects (i.e., caffeine, CGA, cafestol, kahweol, trigonelline, caffeic and ferulic acids). Caffeine and CGA exert an effect on insulin and glucose homeostasis by modulating adenosine receptor signaling, suppressing intestinal glucose absorption (by enhanced generation of gastric inhibitory peptide-1 [GIP-1] and glucagon-like-peptide-1 [GLP-1] or glucose-6-phosphate translocase 1 inhibition), reducing glucose output in the liver (by glucose-6-phosphatase suppression) and by improving secretion of pancreatic islet insulin or peripheral insulin sensitivity and glucose uptake (by glucose transporter type 4 [GLUT4] stimulation and modulation of intracellular signaling pathway activation—Akt, AMPK, MAPK) [68].

Regular coffee intake can lower the levels of proinflammatory biomarkers (such as L-1β, IL-6, TNF-α, C-reactive protein, monocyte chemotactic protein 1, vascular cell adhesion molecule 1, C-peptides, endothelial-leukocyte adhesion molecule 1 and interleukin 18 [IL-18]) in healthy, obese and T2D-affected individuals. However, it may increase the levels of anti-inflammatory adiponectin, interleukin 4 and interleukin 10 [69,70].

## 8. Physical Activity

Physical activity improves glucose control in T2DM, decreases the risk of cardiovascular disorders, contributes to weight loss and improves wellbeing [71,72]. Regular exercise can prevent or delay the development of T2DM [73] and can also provide considerable health benefits for patients with T1DM (e.g., improving cardiovascular efficiency, muscle strength, sensitivity to insulin, etc.) [74]. Recommendations for physical activity and exercise should always be individually tailored [75].

Aerobic exercise is based on repeated and continuous movement of large muscle groups. The aerobic energy-producing systems are involved in such activities as walking, jogging, cycling and swimming. Aerobic training increases mitochondrial density, insulin sensitivity and oxidative enzymes. It enhances compliance and vascular reactivity and improves lung and immune functions and cardiac output [76]. Its volumes, ranging from moderate to high, contribute to markedly lower cardiovascular and overall mortality hazards in T1DM and T2DM [77]. In patients with T1DM, aerobic training enhances cardiac and respiratory fitness, reduces insulin resistance and promotes improvement in lipid levels and endothelial function. In T2DM, regular training decreases A1C, triglycerides, blood pressure and insulin resistance. In turn, high-intensity interval training (HIIT) enhances the oxidative capacity of the skeletal muscles, insulin sensitivity and glycemic control in adults suffering from T2DM [75].

Resistance (strength) training involves physical exercise with free weights, weight machines, body weight or elastic resistance bands. This type of training offers benefits for T2DM individuals, such as improvements in glycemic control, insulin resistance, fat mass, blood pressure, strength and lean body mass. Resistance exercise may reduce the risk of exercise-induced hypoglycemia in type 1 diabetes [75,78].

Flexibility and balance exercises may be of help in elderly diabetics who frequently complain of limited joint mobility, partly due to the formation of advanced glycation end products that are found to accumulate during physiological aging and are intensified by hyperglycemia [75,79].

The advantages of some alternative exercise types, such as yoga and tai chi, are not well established. However, yoga is likely to improve glycemic control, lipid levels, and body composition in T2DM adults [75].

Increasing physical activity reduces the risk of developing diabetic retinopathy [80,81]. Kuwata et al. showed that higher levels of physical activity were independently associated with a lower incidence of DR in patients with type 2 diabetes [82]. The risk of DR progression could be reduced by 40% when the physical activity is for no less than 30 min for five days per week [83].

It has been shown that DM patients leading a sedentary lifestyle have a higher risk of diabetic retinopathy as compared to those living actively [84]. Less physically active diabetic patients showed increased blood flow in the retina on exertion [85,86].

A meta-analysis by Umpierre et al. showed that more structured training, following the ADA’s guideline (>150 min per week) and receiving PA advice alone were correlated with a greater reduction in HbA1c in T2DM patients [87].

A meta-analysis by Boniol et al. also suggested a possible mechanism of PA’s impact on DR due to glycemic control improvement [88]. An alteration in the 25-hydroxyvitamin D (25OH-D) level could be another likely mechanism. Substantial evidence has shown that a higher PA level contributes to an improved 25OH-D status in people of all ages [89,90,91,92,93]. Keech et al. observed lower blood 25OH-D concentration in relation to a higher risk of macrovascular and microvascular events including DR [38].

It has been proved that physical exercise is able to modulate oxidative stress [94]. A few experiments have revealed a decrease in oxidative stress in the retinas of DR mice during physical exercise [95,96,97,98] and positive alterations in microglia in rats with streptozotocin-induced DM following treadmill exercise [99].

It has to be remembered, however, that patients suffering from proliferative diabetic retinopathy should avoid high-intensity aerobic and resistance exercise to reduce the risk of vitreous hemorrhage or retinal detachment [100,101]. All types of physical exercise that may predispose one to elevated systolic blood pressure (Valsalva maneuvers) increase the risk of vitreous hemorrhage [102,103].

Research using animal models of diabetes has shown that resistance exercise can cause increased muscle mass [104]. The skeletal muscle is a substantial glucose reservoir in the body and physical exercise is a potent stimulant of glucose uptake partly via the skeletal muscle glucose transporter protein action [105]. That is why resistance training exerting a direct effect on skeletal muscle is likely to play a role in the management of D2T patients [106].

Resistance training is a good alternative to aerobic exercise. This type of training has not been found to cause a greater number of undesirable events as compared to other types of exercise [106]. Pollock et al. reported that there were more complications during jogging and walking than in strength training in the elderly [107].

Compliance with recommendations related to aerobic exercise can be difficult. Aerobic training should be conducted almost every day or even throughout the week to be effective. However, progressive resistance exercise can be effective if performed only three times a week [106].

Moreover, some coexisting diseases in diabetic patients (cardiovascular and peripheral vascular disorders, neuropathy and motor impairment—foot ulceration, claudication and a risk of falling down) may hinder aerobic performance. In all these situations, resistance training is not only a real alternative but can also be more viable than aerobic exercise. In patients who can safely take aerobic exercise, a combined training program is more effective than the progressive resistance training program alone [106]. Sigal et al. reported that the levels of glycosylated hemoglobin decreased markedly in the combined training group as compared to the only aerobic or resistance training groups [108].

Eight weeks with two or three 45 min. sessions of progressive training are sufficient to improve glycemic control [106].

A reduction in glycosylated hemoglobin by 1% is associated with a 37% decrease in the risk of microvascular complications and a 21% decrease in the risk of diabetes-related death [109].

Snowling and Hopkins observed a 0.5% reduction in glycosylated hemoglobin after progressive resistance training [110].

## 9. Diet and DR

Studies on antioxidants, especially vitamins C and E and carotenoids, as protective factors in an ordinary diet are ambiguous [111].

The prevention and treatment of chronic diseases have to involve limitation of the overall consumption of fat (<30% of energy), saturated fatty acids (<10% of energy) and trans unsaturated fatty acid isomers. As revealed by epidemiological research, high intake of these fats is positively correlated not only with the risk of diabetes but also ischemic heart disease and cancers [112].

Saturated and trans fatty acids can accelerate the development of atherosclerosis by increasing the levels of total cholesterol and LDL cholesterol (LDL-C) and due to proinflammatory and prothrombotic actions. Types of fatty acids and food products containing them have been presented in Table 1 [112].

Fatty products of animal origin are also the source of cholesterol, whose intake should be reduced to 300 mg daily and even to 200 mg daily in the prevention of cardiovascular diseases [112]. The main dietary sources of cholesterol include vitelline, offal, cold meat, pâté and liver. Animal fats should be replaced by vegetable fats, being a source of unsaturated fatty acids. Both monounsaturated and polyunsaturated n-6 fatty acids decrease the level of total cholesterol and LDL-C and increase HDL cholesterol. Fish consumption at least twice a week is extremely important for a proper diet. N-3 polyunsaturated fatty acids contained in fish, such as eicosapentaenoic acid (EPA) and docosahexaenoic acid (DHA), decrease the level of triglycerides [112].

Studies on the effect of fat consumption on the development of diabetic retinopathy are divided. No direct correlation was found between the intake of fat, trans fat, total saturated fatty acid (SFA) and diabetic retinopathy [113,114].

Sasaki et al. failed to find the impact of MUFA on retinopathy [113]. However, Alcubierre et al. reported an inverse correlation of MUFA and oleic acid with the odds of retinopathy [114].

Some evidence indicates that the intake of PUFA may contribute to the prevention of retinopathy [113]. However, one study found no such correlations [114].

Considering fiber intake, two studies observed no association with retinopathy, and one Indian study found an inverse correlation [114,115].

Consuming fish may inhibit the development of retinopathy. The intake of oily fish at least twice a week (as compared to more rare consumption) was associated with a nearly 60% reduction in the risk of retinopathy [116]. Another study reported that 85–141 g of dark fish (salmon, mackerel, swordfish, sardines, bluefish) consumed weekly versus never was related to an almost 70% reduced likelihood of retinopathy [117]. However, 85–141 g of ‘‘other fish’’ (cod, perch, catfish) eaten every week was not associated with retinopathy [117].

Fruit and vegetables constitute an important source of vitamins, minerals, fiber and flavonoids. They should be consumed with each meal in an adequate quantity of at least 400 g daily. Any additional portions of fruit and vegetables in a diet reduces the risk of cardiovascular incidents by 4% and cerebral stroke by 5% [118].

It is believed that extracellular accumulation of glutamate and oxidative stress are the major mechanisms in the process of retinal nerve cell injury [119].

Patients with DR often show deficiency of essential vitamins and minerals, which leads to elevated levels of homocysteine and oxidative stress. Increased serum homocysteine level (Hcy) enhances the risk of microvascular damage [120,121]. A reduction in serum Hcy and in oxidative stress of the mitochondria and cell membranes decreases ischemia and retinal injury [122,123].

## 10. Vitamin A and Carotenoids

Vitamin A (retinol) is a group of fat-soluble retinoids of animal origin, indispensable for the growth and differentiation of cells, immunity and visual processes. In the eye, vitamin A is a component of rhodopsin, a photosensitive pigment. Lutein and zeaxanthin are water-soluble carotenoids of plant origin that easily cross the blood–brain and blood–retina barriers [124]. Concentrated in the macula, they act as strong antioxidants that stabilize cell membranes and protect from oxidative stress. Brazionisa et al. stated that, like in the case of AMD, higher levels of lutein and zeaxanthin were associated with markedly lower DR risk [125]. A randomized study on a few antioxidants revealed that lutein can delay DR progression within 5 years [126]. The application of 10 mg lutein daily improved sensitivity to contrast, glare and visual acuity in patients with nonproliferative DR [127].

## 11. Group B Vitamins

Vitamin B1 (thiamine) is a potent free radical scavenger regulating intracellular glucose and preventing activation of the polyol pathway that is induced by high levels of intracellular glucose [128,129]. Hyperglycemia-induced dysfunction of the polyol pathway is thought to cause DR in rats and humans [130,131]. High serum levels of thiamine protect the vascular endothelium from damage due to advanced glycation end products. Thiamine supplementation in high doses (50–100 mg/day) is safe and useful for neuroprotection, treatment and prevention of end-organ injuries, including DR and diabetic nephropathy [120].

B2 (riboflavin) supplementation in humans increases the synthesis of L-methylofolate, reduces Hcy and lowers blood pressure [120]. In the mouse model, supplementation with riboflavin enhances glucose uptake and alleviates oxidative stress. Therefore, it seems that riboflavin supplementation protects the retina against oxidative stress, hyperglycemia and Hcy-induced injuries [128,132]. Additionally, vitamin B6 and B12 supplementation exerts a beneficial effect as these vitamins decrease the levels of homocysteine.

However, the administration of high doses of vitamin B3 (niacin) can cause or enhance diabetes and increase the risk of cystoid macular degeneration [120].

## 12. Vitamin C

Vitamin C is soluble in water and necessary for the regeneration of other antioxidants, such as vitamin E and glutathione [133,134]. It reduces blood pressure in patients with primary hypertension [135]. Investigations conducted on humans and animals suffering from diabetes showed that oral vitamin C decreases dysfunction of the capillary endothelium [136]. Patients with proliferative DR have a 10-fold lower level of ascorbate in the vitreous humor and a greater tendency for diabetic macular edema [137]. Vitamin C taken with statins decreases nonproliferative DR, in a dose-dependent manner, more than statins alone [138].

## 13. Vitamin D

A proper amount of vitamin D is indispensable for insulin release, sensitivity to insulin, reduction in inflammation and arterial stiffness [120]. It has been shown lately that an optimum level of vitamin D is essential to reduce the risk and severity of DR [139]. Vitamin D plays a role in the functioning of the pancreatic β cells [140]. Its deficiency lowers sensitivity to insulin and elevates the risk of atherosclerosis, CVD, T2DM and hypertension. Clinical studies have shown substantial improvements in sensitivity to insulin and HbA1c when patients received vitamin D3. Its deficiency is associated with T1DM and T2DM [120,141,142,143].

## 14. Vitamin E

Vitamin E supplementation moderately decreases blood pressure, especially the systolic pressure [120]. A study conducted in the Joslin Institute showed that, in patients with T1DM of less than 10 years duration, vitamin E supplementation by a dose of 1800 IU daily improved blood flow in the retina [144]. Oxidative stress, which is elevated in DR, is reduced after treatment with vitamin E [145]. Vitamin E appears to be even more beneficial when it is administered with vitamin C [146].

## 15. Zinc

Zinc is an indispensable cofactor of cell division, DNA synthesis, immune function and carbohydrate and protein metabolism. Zinc deficiency is associated with the progression of chronic pathological conditions, such as metabolic syndrome, diabetes, diabetic microvascular complications and DR [147,148,149]. Low serum levels of zinc are correlated with diabetes duration, elevated HbA1c, hypertension and microcirculatory complications. Zinc level in the serum drops gradually with DR duration and severity [149].

The role of natural medicine in the prevention and inhibition of diabetic retinopathy has become the focus of increasing attention lately. Therapeutic properties can be attributed to a number of herbs [150], including *Litsea japonica*. Its chemical composition shows numerous lactones, alkaloids, essential oils, fatty acids and terpenoids. Treatment with *L. japonica* extract inhibits diabetes-induced blood–retina barrier damage and reduces the expression of vascular endothelial growth factor (VEGF) in mice. This extract also suppresses the degradation of occludin, a major protein within the blood–retina barrier, and can therefore be administered in retinal vascular permeability diseases [151]. Kim et al. examined a soothing effect of the ethanol extract of *L. japonica* on diabetes-induced apoptosis of the retinal neurons in mice. Treatment of mice with *L. japonica* ethanol extract (100 or 250 mg/kg body mass) once daily for 12 weeks caused a slight decrease in blood glucose level, with no significant effect on the level of HbA1c [151].

Additionally, *Ginkgo biloba* shows therapeutic potential thanks to quercitin, which is one of the strongest and most investigated dietary flavonoids, frequently found in seeds, bark, flowers, tea, cruciferous vegetables, onions, apples, berries and nuts [152]. Quercitin suppresses platelet aggregation, capillary permeability and lipid peroxidation. Its bioactivity involves, among others, inhibition of vascular and retinal angiogenesis [153,154] and prevention of oxidative injury to retinal pigment epithelial cells. Quercitin can play a protective role in DR [150].

*Lonicera japonica* has been found to possess a number of biological properties, including antiviral, anti-inflammatory, antibacterial and antipyretic potentials; it also reduces blood lipid levels [155]. Chlorogenic acid, one of the major components of *L. Japonica* [156], derives from quinic acid and caffeic acid and belongs to the class of polyphenols commonly found in beans, potatoes, apples and coffee. Chlorogenic acid may hinder glucose absorption in the small intestine and reduce glucose production in the liver [157]. Moreover, it can prevent glucose intolerance and insulin resistance.

There are many other herbs (*Pueraria lobata*, *Andrographis paniculata*, *Astragalus membranaceus*, *Salvia miltiorrhiza*, *Dendrobium chrysotoxum Vaccinium myrtillus*) known to exert a beneficial effect on the course of diabetic retinopathy [150].

Fiber is another important dietary component found in legumes, wholegrain cereals, fruit and vegetables [158].

Taking all this into account, the Mediterranean diet can be considered the reference diet. Clinical studies and observations performed among inhabitants of the Mediterranean countries have provided evidence that the Mediterranean diet can be accepted as a model pattern of nutrition in the prevention of many diseases, especially cardiovascular disorders, diabetes and obesity [159,160].

Diaz-Lopez et al. studied the Mediterranean diet enhanced with extra virgin olive oil or nuts and compared it with a low-fat diet in more than 3600 participants in a prospective 6-year study. The Mediterranean diet enhanced with olive oil was associated with an over 40% reduced risk of retinopathy [111]. The Mediterranean diet enriched with nuts was correlated with a 37% reduced though statistically insignificant decrease in the risk of retinopathy [111].

Staple foods in this diet include cereals, fruit and vegetables, olive oil, fish, poultry and nuts. Olive oil is known for its high content of monounsaturated fatty acids. This type of fat does not increase cholesterol level. Food products used in the Mediterranean diet are beneficial for reducing inflammation and oxidative stress and help decrease insulin resistance and secretion, these being the pathogenic factors in diabetes [161] and diabetic retinopathy [162,163]. Many vegetables, fruits and seeds contain minerals, polyphenols and other phytochemicals which alleviate oxidative stress, inflammatory conditions and insulin resistance [164]. In fact, high consumption of fruit and vegetables rich in flavonoids is correlated with a lower risk of diabetic retinopathy [111].

## 16. Oxidative Stress and Antioxidant Supplementation

Oxidative stress has been shown to play a key role in DR pathophysiology. It has also been reported that antioxidant supplementation is recommended in adults with T1DM and T2DM showing no features of retinopathy or with symptoms of mild to moderate nonproliferative DR free of diabetic macular edema [25]. Antioxidants are defined as compounds able to delay, inhibit or prevent oxidative lesions. They can be divided into the enzymatic and non-enzymatic ones. The former obtain their antioxidant activity via the breakdown and removal of free radicals. They include intracytosolic enzymes (catalase, superoxide dismutase, glutathione peroxidase and peroxiredoxin), which carry chemical reactions in the presence of a few cofactors, including coenzyme Q10 (ubichinon), copper (Cu), manganese (Mn), zinc (Zn) or selenium (Se) [25,165,166,167]. The nonenzymatic antioxidants disturb the chain reactions of free radicals. They include vitamins C, E and A and polyphenols, and most of them can be extracted from natural sources (fruits and vegetables) [25,165,168].

Polyphenols appear to decrease DR progression in animals [169]. However, their role in the clinical treatment of DR patients is not well known. Polyphenols are the most abundant antioxidants in our diet [170]. Natural polyphenols, the secondary plant metabolites, are found in fruit, vegetables, wholegrain foods and their products, e.g., chocolate, wine, olive oil or tea.

Resweratrol (3,4′,5-trihydroksystytilben; RESV) is the best studied polyphenol. Oral administration of RESV in diabetic rats alleviates oxidative stress and reduces the level of inflammatory cytokines [171,172,173,174]. Pterostilben is a natural structural analog of resweratrol showing beneficial biological action that exceeds that of resweratrol [175]. This polyphenol shows high efficacy against various pathological effects of diabetes. Pterostilben reduces glucose levels in the blood of rats with hyperglycemia [38] and improves carbohydrate metabolism [176].

Chlorogenic acid (CGA) is a polyphenol found in a variety of products, such as coffee, grains, potatoes and apples. It is the ester of caffeic and quinic acids, showing antibacterial, anti-inflammatory, antioxidant and antineoplastic effects [177,178,179]. CGA can delay glucose absorption in the small intestine and decrease glucose production in the liver [180,181]. It is believed that CGA stimulates the secretion of glucagon-like peptide 1 known to exert a beneficial effect on the response to glucose in pancreatic beta cells [179]. In the liver, CGA inhibits glucose-6-phospatase, thus decreasing hepatic glucose production [157,181]. CGA reduces the hyperpermeability of retinal vessels in diabetic rats. The diabetic rats with higher levels of VEGF and down-regulation of occludin, claudin-5 and ZO-1 showed breakdown of the blood–retina barrier and intensified vascular leakage. CGA managed to maintain occludin expression and reduced the levels of VEGF, which decreased the BRB breakdown and inhibited vascular leakage. Thus, CGA may prevent BRB breakdown in retinopathy [182].

It has been found that n3-PUFA can reduce the number of retinal acellular capillaries in diabetes and inflammatory markers in the retina of diabetic animals [183,184].

Diabetic patients taking n-3 PUFA supplements for 18 months had a lower risk of retinopathy [185].

The effect of vitamin C intake on DR is not certain [115,183,184,185,186,187]. The use of vitamin C supplements may suggest that this vitamin is involved in retinopathy prevention [115].

Millen et al. [186], conducting a prospective study of the ARIC participants, failed to observe an effect of vitamin C and E intake from food alone or a combination of food and supplements on the risk of retinopathy. However, they found an interaction between race and vitamin E; high consumption of vitamin E from food alone, or food combined with supplements, was associated with a higher prevalence of retinopathy in Caucasians but not in African Americans [186]. A diminished likelihood of retinopathy was noted in those taking (for 3 years) vitamin C, E and multivitamin supplements in comparison with non-users [186]. However, in a cross-sectional analysis of NHANES III participants, Millen et al. found no relationship between taking vitamin C or E supplements for a long time (for 5 vs. 1 year) and the presence of retinopathy [188].

The intake of vitamin D and fish oil supplements was not associated with the risk of retinopathy in the ARIC [117].

Lutein intake was not significantly correlated with retinopathy [33]. In one study, the serum level of lutein was found to be decreased in patients with nonproliferative diabetic retinopathy as compared to diabetics without retinopathy [189].

Another study suggested a positive impact of high plasma levels of lutein/zeaxanthin and lycopene on the risk of diabetic retinopathy [125]. However, increased consumption of b-carotene was found to be related to a lower risk and was not associated with retinopathy [115,187].

Calcium dobesilate (CaD) is considered to be an angioprotective drug. For many years CaD was used in diabetic retinopathy [190,191]. Researchers investigating its efficacy are divided. A few clinical studies showed delayed progression of diabetic retinopathy after long-term oral treatment with CaD. It was found to inhibit changes in tight junction proteins, ICAM-1 and leukocyte adhesion to the retinal vessels which are known to lie at the base of growing permeability of the blood–retina barrier. These results were correlated with the inhibition of oxidative/nitrosative stress and the activity of p38 MAPK and NF-κB [192]. A positive effect of CaD (2000 mg/day for 2 years) was found in patients suffering from early diabetic retinopathy [193]. Other studies failed to show a positive effect of CaD on the retinas of diabetic patients [194,195]. In a recent study (CALDIRET) (30), in which patients with mild to moderate non-proliferative diabetic retinopathy were followed up with for 5 years, CaD did not reduce diabetic macular edema [196].

## 17. Discussion

Summing up, the prevention of diabetic retinopathy should involve adequate control of glycemia, lipidogram and blood pressure, which can be obtained not only by pharmacological treatment but first of all by promoting a healthy lifestyle, changes in dietary habits and increased physical activity. The risk of DR progression could be reduced by 40% when physical activity is undertaken for no less than 30 min for five days per week [83]. In individuals who increased their physical activity and changed their eating habits, decreases were observed in the occurrence of diabetes and its vascular complications. It is very important to educate patients [197]. Further research on diabetic retinopathy is necessary.

## Figures and Tables

**Table 1 nutrients-14-01252-t001:** Sources of fatty acids.

Types of Fatty Acids	Products
Saturated	Dairy products, whole-fat (butter, cheese, cream, milk), lard, tallow, fat meat, palm oil, coconut oil
Monounsaturated	Olive oil, rapeseed oil, margarine, almonds, hazelnuts, tuna, sardines
Polyunsaturated	–omega 6: corn oil, soya oil, sunflower oil, walnuts, margarines –omega 3: green leaves, seeds, linseed oil, rapeseed oil, soya oil, fish oils (cod, mackerel, salmon)
“Trans” (from hydrated oils)	margarines, confectionary fats (crackers, cakes, doughs), “fast-food”

## Data Availability

Not applicable.

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
