# Peer review of "The Effect of Diet and Lifestyle on the Course of Diabetic Retinopathy—A Review of the Literature"

_nutrients, 2022, doi:10.3390/nu14061252_

Round 1

Reviewer 1 Report

  1. This is a narrative review on the relationship between lifestyle factors, especially exercise and diet, and diabetic retinopathy. I read the paper with great interest.
  2. My major concern is that role of physical activity and diet in diabetic retinopathy has been already described (e.g., https://doi.org/10.1007/s00592-019-01319-4 https://doi.org/10.1159/000502387 https://doi.org/10.1007/s10654-017-0338-8 https://doi.org/10.1007/s11892-013-0384-x ). It would be nice if the authors could show the rationale or research gap to write the present narrative review using existing literature. Clear descriptions of novelty of originality of the present narrative review may help readers to understand the importance of the present paper. Following reporting guidelines or checklists related to narrative reviews may help writing. https://www.equator-network.org/reporting-guidelines/rameses-publication-standards-meta-narrative-reviews/ https://www.elsevier.com/__data/promis_misc/ANDJ%20Narrative%20Review%20Checklist.pdf
  3. Another point is the balance of contents of the present paper. In my impression, explanations about exercise and diet are very concise; the main text does not match with the title and abstract. Readers may expect more details on pathophysiology, current evidence (if systematic review was employed, much better) for exercise and diet.
  4. Relating to the previous comment, selection of papers is challenging in writing narrative review. It would be nice if clarified why and how the authors selected the papers cited in the present paper.
  5. Please specify study design in the title or abstract. This information helps readers to understand the structure of the paper.
  6. I could not find an explanation about conflict of interest of the authors. Please clarify.

Author Response

Response to Reviewer 1

  1. Thank you very much for your important comments. I'm happy to use them to make the manuscript more interesting.
  2. Thank you for the links to the articles. We read them with great interest and used them in our manuscript.
  3. We have added information about diet and exercise
  4. Thank you for your question. Diabetic retinopathy is a very important social problem. It is very important to determine all the factors affecting its course. We tried to write our manuscrypt by reviewing the latest scientific literature.
  5. The missing elements are completed in the title and summary.

Title - The effect of diet and lifestyle on the course of diabetic retinopathy – a review of literature

Abstract - In our work, we present a review of literature to show that physical exercise and adequate diet can significantly reduce the risk of diabetes and diabetic retinopathy.

  1. Thank you for your attention. We have added an explanation about conflict of interest of the authors.

Authors do not report a conflict of interest related to this article

Reviewer 2 Report

Dear Editor,

I carefully read the manuscript by Bryl et al.

My comments and suggestions for the authors are the following:

  • English language is good quality. However, there are some typos that need to be corrected.
  • Lines 115-117: This definition of "hyperlipidemia" is formally uncorrect. The authors should more appropriately refer to the definition of the ESC/EAS guidelines. Moreover, they should differentiate between different forms of hyperlipemia.
  • Line 127: I suppose that the authors used "fenofibrat" as referring to "fenofibrate". The language mistake should be ammended.
  • I warmly suggest to the authors to add another paragraph entitled "Discussion" in which comment the evidence degree of the effect of diet and lifestyle on the course of diabetic retinopathy. The authors should also discuss the strenghts and limitations of this approach. Are more research clincial studies needed? This and other points should be more detailed in the manuscript.
  • References have not been formatted following the Instructions for the Authors of the Journal.
  • In the background, the authors should include more information as regard the awareness of diabetes in population. They should appropriately refer to doi: 10.1016/j.numecd.2020.03.005.

Author Response

Thank you very much for your important comments. We will use them to make the manuscript more interesting

  1. Language mistakes were corrected by English native speaker from the Foreign Language Study of the Medical University in Bialystok.
  2. Thank you very much for your suggestions. The definition of hyperlipidemia has been changed.

Definition describes hyperlipidemia as low-density lipoprotein (LDL), total cholesterol, triglyceride levels, or lipoprotein levels greater than the 90th percentile in comparison to the general population, or an HDL level less than the 10th percentile when compared to the general population. Lipids typically include cholesterol levels, lipoproteins, chylomicrons, VLDL, LDL, apolipoproteins, and HDL.

  1. Entitled "Discussion"  has been added. Diabetic retinopathy is a very important social problem. There are a lot of studies available on the prevention and treatment of diabetic complications.  However, further research are needed.

Discussion

Summing up, the prevention of diabetic retinopathy should involve adequate control of glycemia, lipidogram and blood pressure, which can be obtained not only by pharmacological treatment but first of all by promoting healthy lifestyle, changes in dietary habits and increased physical activity. The risk of DR progression could be reduced by 40% when physical activity is  no less than 30 minutes for five days per week [71]. In individuals who had increased their physical activity and changed eating habits a decrease was observed in the occurrence of diabetes and its vascular complication. It is very important to educate patients [132]. Further research on diabetic rethionopathy are necessary.

  1. Thank you for your attention. References have been improved following the Instructions for the Authors of the Journal.

Forbes, J.M.; Cooper, M.E. Mechanisms of diabetic complications. Physiol Rev 2013, 93, 137–188.

  1. Thank you very much for your suggestions. DOI: 10.1016/j.numecd.2020.03.005. We have added to our manuscript
  2. Cicero, A.F.G.; Fogacci, F.; Tocci, G, et al. Awareness of major cardiovascular risk factors and its relationship with markers of vascular aging: data from the Brisighella Heart Study. Nutr Metab Cardiovasc Dis. 2020, 30(6), 907–914. doi:10.1016/j.numecd.2020.03.005. Epub 2020 Mar 16. PMID: 32249143.

Reviewer 3 Report

The authors took an interesting topic (diet/ lifestyle and risk of diabetic retinopathy) treated this topic superficially. Unfortunately, the content of the different sections barely touches the link of diet and lifestyle with the risk of diabetic retinopathy.

But the content has nothing to do with the title and the aim.

Here are few of my comments:

  1. The mechanism by which glycemia, high BP and dyslipidemia increase the risk of diabetic retinopathy should be briefly touched.
  2. Sections HbA1c and metabolic memory should be combined as they are dealing with the same subject – glycemic control.
  3. Section 8. What about existing evidence on the effect of coffee/caffeine and tea consumption and risk of diabetic retinopathy?
  4. Section 9. The effect of different types of physical exercise on glycemia and other risk factors are presented briefly in the manuscript. However, nothing is shown about the existing studies between physical exercise and diabetic retinopathy. And plenty of studies are available.
  5. Section 9. Resistance training does not promote per se improved glycemic control or weight loss neither in type 1 not in type 2 diabetes. However, this type of exercise is recommended. Its short- and long-term effect on glycemia should be discussed and put in the context of the risk of DR. Also, existing research on this topic should be presented and discussed. This would be expected given the title and the aim of this review.
  6. Section 10. When it comes about the diet, the authors nicely present sources of each type of fats and briefly state the risk of diabetes and CV diseases. What about the dietary patterns and eventually the association of different type of macro/micronutrients and diabetic retinopathy? Again, this is the aim of this manuscript! There is such mention only for vitamin D. What about epidemiological studies and proven or presumed mechanisms for the remaining ones?

The authors should have performed a careful review of the existing literature on this subject, as studies are available, but they are not so many. I invite the authors to read the following paper on this subject: Dow C, Mancini F, Rajaobelina K, Boutron-Ruault MC, Balkau B, Bonnet F, Fagherazzi G. Diet and risk of diabetic retinopathy: a systematic review. Eur J Epidemiol. 2018 Feb;33(2):141-156.

  1. Section 10. First sentence. This is a superficial statement. I agree with the authors that in general studies on the effect of antioxidants on human health may provide contradictory results. Again, here the authors should discuss the available evidence relating the use of food supplements and diabetic retinopathy instead of dismissing is briefly.
  2. Section 11. There is no recommendation for routine supplementation with antioxidants or other food supplements in diabetes. If there are studies on the effect of antioxidants use on the risk of diabetic retinopathy or its evolution, these should be presented and discussed. Not only a brief statement on polyphenols.
  3. I do not understand the purpose of section 12 in this manuscript that should discuss the link of diet/lifestyle and diabetic retinopathy.

Author Response

Thank you very much for your important comments. We will use them to make the manuscript more interesting

  1. Thank you very much for your suggestions. Sections HbA1c and metabolic memory are together.

Hemoglobin A1C (HbA1C< 7% ) and metabolic memory 

  1. Thank you for your attention. We have corrected section 9.

Increasing physical activity reduces the risk of developing diabetic retinopathy [68,69]. Kuwata et al. showed that higher levels of physical activity were independently associated with a lower incidence of DR in patients with type 2 diabetes [70]. The risk of DR progression could be reduced by 40% when physical activity is  no less than 30 minutes for five days per week [71].

  1. Praidou, A.; Harris, M.; Niakas, D,; Labiris,G. J. Physical activity and its correlation to diabetic retinopathy. Diabetes Complications. 2017, 31(2), 456-461. doi: 10.1016/j.jdiacomp.2016.06.027. Epub 2016 Jun 29.
  2. Yan, X.; Han, X.; Wu, Ch.; Shang, X.; Zhang, L.; He, M. Effect of physical activity on reducing the risk of diabetic retinopathy progression: 10-year prospective findings from the 45 and Up Study. PLoS One. 2021 Jan 14, 16(1), e0239214. doi: 10.1371/journal.pone.0239214. eCollection 2021.
  3. H, Okamura S, Hayashino Y, Tsujii S, Ishii H, Diabetes D, et al. Higher levels of physical activity are independently associated with a lower incidence of diabetic retinopathy in Japanese patients with type 2 diabetes: A prospective cohort study, Diabetes Distress and Care Registry at Tenri (DDCRT15). PLoS One. 2017, 12(3), e0172890 Epub 2017/03/04. 10.1371/journal.pone.0172890
  4. Dirani, M.; Crowston, J.; van Wijngaarden, P. Physical inactivity as a risk factor for diabetic retinopathy?. A review. Clin Experiment Ophthalmol. 2014, 42(6), 574–581. 10.1111/ceo.12306
  5. Section 10 - we have added more information.

Patients with DR often show deficiency of essential vitamins and minerals, which leads to elevated levels of homocysteine and oxidative stress. Increased serum homocysteine level (Hcy) enhances the risk of microvascular damage  [72,73]. A reduction in serum Hcy and in oxidative stress of the mitochondria and cell membranes decreases ischemia and retinal injury [74,75].

Vitamin A and carotenoids

Vitamin A (retinol) is a group of fat-soluble retinoids of animal origin, indispensable for the growth and differentiation of cells, immunity and visual processes.  In the eye, vitamin A is a component of rhodopsin, a photosensitive pigment. Lutein and zeaxanthin are water-soluble carotenoids of plant origin that easily cross the blood-brain and blood-retina barriers [76]. Concentrated in the macula, they act as strong antioxidants that stabilize cell membranes and protect from oxidative stress. Brazionisa et al. stated that like in the case of AMD, higher levels of lutein and zeaxanthin were associated with markedly lower DR risk [77]. A randomized study on a few antioxidants revealed that lutein can delay DR progression within 5 years [78]. The application of 10 mg lutein daily improved sensitivity to contrast, glare and visual acuity in patients with nonproliferative DR [79].

Group B vitamins

Vitamin B1 (thiamine) is a potent free radical scavenger regulating intracellular glucose and preventing activation of the polyol pathway that is induced by high level of intracellular glucose [80,81]. Hyperglycemia-induced dysfunction of the polyol pathway is thought to cause DR in rats and humans [82,83]. High serum levels of thiamine protect vascular endothelium from damage to advanced glycation end products. Thiamine supplementation in high doses (50-100 mg/day) is safe and useful for neuroprotection, treatment and prevention of end-organ injuries, including DR and diabetic nephropathy [72].

B2 (riboflavin) supplementation in humans increases the synthesis of L-methylofolate, reduces Hcy and lowers blood pressure [72]. In the mouse model, supplementation with riboflavin enhances glucose uptake and alleviates oxidative stress. Therefore, it seems that riboflavin supplementation protects the retina against oxidative stress, hyperglycemia and Hcy-induced injuries [84]. Also vitamin B6 and B12 supplementation exerts a beneficial effect as these vitamins decrease the levels of homocysteine. 

However, the administration of high doses of vitamin B3 (niacin) can cause or enhance diabetes and increase the risk of cystoid macular degeneration [72].

Vitamin C

Vitamin C is soluble in water and necessary for the regeneration of other antioxidants, such as vitamin E and glutathione [85,86]. It reduces blood pressure in patients with primary hypertension [87]. Investigations conducted on humans and animals suffering from diabetes showed that oral vitamin C decreases dysfunction of capillary endothelium [88]. Patients with proliferative DR have a 10-fold lower level of ascorbate in the vitreous humor and greater tendency for diabetic macular edema [89]. Vitamin C taken with statins decreases nonproliferative DR, in a dose-dependent manner, more than statins alone [90].

Vitamin D

Proper amount of vitamin D is indispensable for insulin release, sensitivity to insulin,  reduction in inflammation and arterial stiffness [72]. It has been shown lately that the optimum level of vitamin D is essential to reduce the risk and severity of DR [91]. Vitamin D plays a role in the functioning of pancreatic β cells [92]. Its deficiency lowers sensitivity to insulin and elevates the risk of atherosclerosis, CVD, T2DM and hypertension. Clinical studies have shown substantial improvement in sensitivity to insulin and HbA1c, when patients received vitamin D3.  Its deficiency is associated with T1DM and T2DM [72,93-95].

Vitamin E

Vitamin E supplementation moderately decreases blood pressure, especially the systolic pressure [72]. The study conducted in Joslin Institute showed that in patients with  T1DM of less than 10 years duration vitamin E supplementation in a dose of 1800 IU daily improved blood flow in the retina [96]. Oxidative stress which is elevated in DR is reduced after treatment with vitamin E [97]. Vitamin E appears to be even more beneficial when it is administered with vitamin C [98].

Zinc

Zinc is an indispensable cofactor of cell division, DNA synthesis, immune function, and carbohydrate and protein metabolism. Zinc deficiency is associated with the progression of chronic pathological conditions, such as metabolic syndrome, diabetes, diabetic microvascular complications and DR [99-101]. Nurses' Health Study revealed a 20% difference in the risk of diabetic morbidity between the highest and lowest quintile of zinc intake [102]. Low serum levels of zinc are correlated with diabetes duration, elevated HbA1c, hypertension and microcirculatory complications. Zinc level in the serum drops gradually with DR duration and severity  [101].

The role of natural medicine in the prevention and inhibition of diabetic retinopathy has become the focus of increasing attention lately. Therapeutic properties can be attributed to a number of herbs [102], including Litsea japonica. Its chemical composition shows numerous lactones, alkaloids, essential oils, fatty acids and terpenoids. Treatment with L. japonica extract inhibits diabetes-induced blood-retina barrier damage and reduces the expression of vascular endothelial growth factor (VEGF) in mice. This extract also suppresses degradation of occludin, a major protein within the blood-retina barrier, and can be therefore administered in retinal vascular permeability diseases [103]. Kim et al. examined a soothing effect of the ethanol extract of L. japonica on diabetes-induced apoptosis of retinal neurons in mice. Treatment of mice with L. japonica ethanol extract (100 or 250 mg/kg body mass) once daily for 12 weeks caused a slight decrease in blood glucose level, with no significant effect on the level of HbA1c [102].

Also Ginkgo biloba shows a therapeutic potential  thanks to quercitin, which is one of the strongest and most investigated dietary flavonoids, frequently found in seeds, bark, flowers, tea, cruciferous vegetables, onions, apples, berries and nuts [104]. Quercitin suppresses platelet aggregation, capillary permeability and lipid peroxidation. Its bioactivity involves, among others, inhibition of vascular and retinal angiogenesis [105,106] and prevention of oxidative injury to retinal pigment epithelial cells. Quercitin can play a protective role in DR [102].

Lonicera japonica has been found to possess a number of biological properties, including antiviral, anti-inflammatory, antibacterial and antipyretic potentials; it also reduces  blood lipid levels [107]. Chlorogenic acid, one of the major components of L. Japonica [108],  derives from quinic acid and caffeic acid, and belongs to the class of polyphenols commonly found in beans, potatoes, apples and coffee. Chlorogenic acid may hinder glucose absorption in the small intestine and reduce glucose production in the liver [109]. Moreover, it can prevent glucose intolerance and insulin resistance.

There are many other herbs (Pueraria lobata, Andrographis paniculata, Astragalus membranaceus, Salvia miltiorrhiza, Dendrobium chrysotoxum Vaccinium myrtillus) known to exert a beneficial effect on the course of diabetic retinopathy [102].

  1. Thank you very much for your suggestions. Dow C, Mancini F, Rajaobelina K, Boutron-Ruault MC, Balkau B, Bonnet F, Fagherazzi G. Diet and risk of diabetic retinopathy: a systematic review. Eur J Epidemiol. 2018 Feb;33(2):141-156. We added to our manuscript.
  2. Section 12 deleted.

Round 2

Reviewer 1 Report

I have no additional comments.

Author Response

Thank you very much for your cooperation. Best regards. Anna Bryl

Reviewer 2 Report

Dear Editor,

I carefully read the revised version of the manuscript that is significantly improved in comparison with the previous one. I warmly recommend its publication in the Journal.

Author Response

Thank you for your cooperation. Best regards. Anna Bryl

Reviewer 3 Report

The manuscript of Bryl et al has been much improved. However not all previous comments have been addressed and neither a rebuttal has been provided for the unadressed comments. Given that the aim of this manuscript is to discuss the effect of lifestyle on diabetic retinopathy (DR), I re-iterate my previous comments that remained unaddressed:

Section 7. The authors should discuss the existing evidence on the effect of coffee/caffeine and tea consumption and risk of diabetic retinopathy?

Section 8. The authors have added 2 studies in a very brief section, while the aim of the manuscript is the effect of lifestyle on the risk of DR. A lot of text describes the overall beneficial effect of physical exercise on CV system and diabetes. However more emphasize is needed on the numerous studies showing the effect physical exercise and diabetic retinopathy.

Please see for some of the existing studies the literature review performed by Ren et al. https://doi.org/10.1007/s00592-019-01319-4

Section 8. Resistance training does not promote per se improved glycemic control or weight loss neither in type 1 not in type 2 diabetes. On the contrary immediate effect is hyperglycemia and increase in body weight. However, this type of exercise is recommended. Its short- and long-term effect on glycemia should be discussed and put in the context of the risk of DR. Also, existing research on this topic should be presented and discussed. This would be expected given the title and the aim of this review. The newly added sentence on the about hypoglycemia does not address the comment as the paper refers to DR.

Section 9. The authors have added a very detailed section on vitamins which is good. However, I will re-state my previous comment which remained mainly unaddressed (with the exception of micronutrients).

When it comes about the diet, the authors nicely present sources of each type of fats and briefly state the risk of diabetes and CV diseases. What about the association of different type of macronutrients (at least about types of dietary fats) and diabetic retinopathy? Again, this is the aim of this manuscript!

Section 11. There is no recommendation for routine supplementation with antioxidants or other food supplements in diabetes according to current diabetes management guidelines. The authors support their statement with a review of the literature. The existing studies on the effect of antioxidants use on the risk of diabetic retinopathy or its evolution, these should be presented and discussed. Not only a brief statement on polyphenols.

Author Response

Response to Rewiever 3

Thank you very much for your important comments.

Section 7.

We have completed the information concerning the effect of coffee and tea intake on the risk of diabetic retinopathy.

It has been shown that tea acts as a potent neuroprotector in the retina [5], preventing formation of acellular capillary vessels and pericyte ghosts in diabetic rats [1]. Green tea can protect diabetic retinal neurons and regulate the subretinal environment by decreasing  ROS generation due to increased expression of glutamate transporter, restored intercellular connections and glutamine/glutamate circulation [2]. Moreover, a low dose of green tea  is likely to improve antioxidant defense, decrease inflammatory markers and prevent thickening of the basement membrane of the retina [3]. Black tea, which lowers blood sugar and thus inhibits pathological biochemical indices, is able to delay diabetic cataract development  [4].

As found in a Chinese case–control study, involving sex and age-matched controls suffering from diabetes but not diabetic retinopathy, regular consumption (every week for not less than 1 year) of Chinese green tea was associated with reduced probability of retinopathy in women, but not in men [6].

  1. Mustata, G.T.; Rosca, M.; Biemel, K.M.; Reihl, O.; Smith, M.A.; Viswanathan, A.; Strauch, C.; Du, Y.; Tang, J.; Kern, T.S. et al. Paradoxical effects of green tea (Camellia Sinensis) and antioxidant vitamins in diabetic rats: Improved retinopathy and renal mitochondrial defects but deterioration of collagen matrix glycoxidation and cross-linking. Diabetes. 2005, 54, 517–526. doi: 10.2337/diabetes.54.2.517. [PubMed] [CrossRef] [Google Scholar]
  2. Silva, K.C.; Rosales, M.A.B.; Hamassaki, D.E.; Saito, K.C.; Faria, A.M.; Ribeiro, P.A.O.; de Faria, J.B.L.; de Faria, J.M.L. Green tea is neuroprotective in diabetic retinopathy. Ophthalmol. Vis. Sci. 2013, 54, 1325–1336. doi: 10.1167/iovs.12-10647. [PubMed] [CrossRef] [Google Scholar]
  3. Kumar, B.; Gupta, S.K.; Nag, T.C.; Srivastava, S.; Saxena R. Green tea prevents hyperglycemia-induced retinal oxidative stress and inflammation in streptozotocin-induced diabetic rats. Ophthalmic Res. 2012, 47, 103–108. doi: 10.1159/000330051. [PubMed] [CrossRef] [Google Scholar]
  4. Vinson, J.A.; Zhang, J. Black and green teas equally inhibit diabetic cataracts in a streptozotocin-induced rat model of diabetes. Agric. Food Chem. 2005, 53, 3710–3713. doi: 10.1021/jf048052l. [PubMed] [CrossRef] [Google Scholar]
  5. Jin-Ming, Meng.; Shi-Yu, Cao.; Xin-Lin Wei, Ren-You Gan, Yuan-Feng Wang, Shu-Xian Cai, Xiao-Yu Xu, Pang-Zhen Zhang, Hua-Bin Li. Effects and Mechanisms of Tea for the Prevention and Management of Diabetes Mellitus and Diabetic Complications: An Updated Review. Antioxidants (Basel) 2019 Jun; 8(6): 170. Published online 2019 Jun 10. doi: 10.3390/antiox8060170.
  6. Ma, Q.; Chen, D.; Sun, H.P.; Yan, N.; Xu, Y.; Pan, C.W. Regular Chinese green tea consumption is protective for diabetic retinopathy: a Clinic-Based Case-Control Study. J Diabetes Res. 2015, 2015, 231570.

Coffee is one of the most consumed beverages worldwide. The main commercial coffee blends are Arabica (Coffea arabica L.) and Robusta (Coffea canephora Pierre ex Froehner) [1]. Its main component is caffeine which belongs to antioxidants and its long-term intake in large quantities reduces oxidative stress [2]. Other components of coffee beans include carbohydrates, proteins, fats, alkaloids, diterpenes, free amino acids, melanoides and minerals, both macro- and microelements [3]. Coffee contains microelements that show antioxidant effects, such as manganese, zinc, copper and iron [1]. It can also be a source of fluorine, chromium and cobalt [1].

A Norwegian cohort study revealed an approximately 35% reduced risk of T2D associated with high (in comparison to low) consumption of brewed coffee and other types of coffee [2]. Likewise, the results of a Finnish cohort study revealed that both the intake of boiled coffee and consumption of drip coffee were inversely correlated with T2D [3].

A number of coffee components have been found to show anti-inflammatory effects (i.e. caffeine, CGA, cafestol, kahweol, trigonelline, caffeic and ferulic acids). Caffeine and CGA exert an effect on insulin and glucose homeostasis by modulating adenosine receptor signaling, suppressing intestinal glucose absorption (by enhanced generation of gastric inhibitory peptide-1 [GIP-1] and glucagon-like-peptide-1 [GLP-1] or glucose-6-phosphate translocase 1 inhibition), reducing glucose output in the liver (by glucose-6-phosphatase suppression), and by improving secretion of pancreatic islet insulin or peripheral insulin sensitivity and glucose uptake (by glucose transporter type 4 [GLUT4] stimulation and modulation of intracellular signaling pathway activation - Akt, AMPK, MAPK) [4].

Regular coffee intake can lower the level of proinflammatory biomarkers, such as  L-1β, IL-6, TNF-α, C-reactive protein, monocyte chemotactic protein 1, vascular cell adhesion molecule 1, C-peptides, endothelial-leukocyte adhesion molecule 1, and interleukin 18 [IL-18]) in healthy, obese and T2D-affected individuals. However, it may increase the levels of anti-inflammatory adiponectin, interleukin 4 and interleukin 10  [5,6].

  1. Hjellvik, V.; Tverdal, A.; Strom, H. Boiled coffee intake and subsequent risk for type 2 diabetes. 2011, 22, 418–421.
  2. Olechno, E.; Pus´cion-Jakubik, A.; Socha, K.; Zujko, M.E. Coffee Infusions: Can They Be a Source of Microelements with Antioxidant Properties? Antioxidants (Basel). 2021, Oct 27, 10(11), doi: 10.3390/antiox10111709
  3. Tuomilehto, J.; Hu, G.; Bidel, Set al. , Coffee consumption and risk of type 2 diabetes mellitus among middle-aged Finnish men and women. JAMA. 2004, 291, 1213–1219.
  4. Carlström, M.; Larsson, S.C. Coffee consumption and reduced risk of developing type 2 diabetes: A systematic review with meta-analysis. Rev. 2018, 76, 395–417. [CrossRef] [PubMed]
  5. Natella, F.; Scaccini, C. Role of coffee in modulation of diabetes risk. Nutr Rev. 2012, 70, 207–217.
  6. Akash, M.S.; Rehman, K.; Chen, S. Effects of coffee on type 2 diabetes mellitus. 2014, 30, 755–763.

Section 9.

We have completed the information.

Diaz-Lopez et al. studied the Mediterranean diet enhanced with extra virgin olive oil or nuts,  and compared it with a low-fat diet in more than 3600 participants in a prospective 6-year study. The Mediterranean diet enhanced with olive oil was associated with over 40% reduced risk of retinopathy [1].  The Mediterranean diet enriched with nuts was correlated with a 37% reduced, though statistically insignificant decrease in the risk of retinopathy [1].

A Japanese cohort study of T2D individuals revealed that high fruit intake (C 173.2 g per day, i.e. an apple or two bananas), was associated with over a 50% reduction in the risk of incident retinopathy, as compared to patients who consumed less than 53.2 g of fruit daily [2].

Milk intake has also been assessed; the consumption of whole milk and skim/low fat milk was not found to be associated with retinopathy in more than 1350 patients with type 2 diabetes [3].

Fish may inhibit the development of retinopathy. The intake of oily fish at least twice a week (as compared to more rare consumption), was associated with a nearly 60% reduction in the risk of retinopathy [4]. Another study reported that 85–141 g of dark fish, (salmon, mackerel, swordfish, sardines, bluefish) consumed weekly versus never was related to almost 70% reduced likelihood of retinopathy [3]. However, 85–141 g of ‘‘other fish’’ (cod, perch, catfish), eaten every week was not associated with retinopathy [3].

The intake of vitamin D and fish oil supplements, versus the non-use, was not related to the risk of retinopathy in the ARIC [3]. In an RCT, diabetics who were randomly given n-3 PUFA supplements for 18 months had a lower risk of retinopathy [5].

Studies on the effect of fat consumption on the development of diabetic retinopathy are divided. No direct correlation was found between the intake of fat, trans fat, total saturated fatty acid (SFA) and diabetic retinopathy [6,7].

Sasaki et al. failed to find the  impact of MUFA on retinopathy [6]. However, Alcubierre et al.  reported an inverse correlation of MUFA and oleic acid with odds of retinopathy [7].

Some evidence indicates that the intake of PUFA may contribute to the prevention of  retinopathy [6]. However, one study found no such correlations [7].

Considering fiber intake, two studies observed no association with retinopathy, and one Indian study found an inverse correlation [2, 7] .

  1. Diaz-Lopez A, Babio N, Martinez-Gonzalez MA, Corella D, Amor AJ, Fito M, et al. Mediterranean diet, retinopathy, nephropathy, and microvascular diabetes complications: a post hoc analysis of a randomized trial. Diabetes Care. 2015, 38(11), 2134–2141.
  2. Tanaka, S.; Yoshimura, Y.; Kawasaki, R.; Kamada, C.; Tanaka, S.; Horikawa, C. et al. Fruit intake and incident diabetic retinopathy with type 2 diabetes. 2013, 24(2), 204–211.
  3. Millen, A.E.; Sahli, M.W.; Nie, J.; LaMonte, M.J.; Lutsey, P.L.; Klein, B.E. et al. Adequate vitamin D status is associated with the reduced odds of prevalent diabetic retinopathy in African Americans and Caucasians. Cardiovasc. 2016, 15(1), 128.
  4. Sala-Vila, A.; Diaz-Lopez, A.; Valls-Pedret, C.; Cofan, M.; GarciaLayana, A.; Lamuela-Raventos, R.M. et al. Dietary Marine omega-3 fatty acids and incident sight-threatening retinopathy in middleaged and older individuals with type 2 diabetes: prospective investigation from the PREDIMED trial. JAMA Ophthalmol. 2016, 134(10), 1142–1149.
  5. Roig-Revert, M.J.; Lleo-Perez, A.; Zanon-Moreno, V.; Vivar-Llopis, B.; Marin-Montiel, J.; Dolz-Marco, R. et al. Enhanced oxidative stress and other potential biomarkers for retinopathy in type 2 diabetics: beneficial effects of the nutraceutic supplements. Biomed Res Int. 2015, 2015, 408180.
  6. Sasaki, M.; Kawasaki, R.; Rogers, S.; Man, R.E.; Itakura, K.; Xie, J. et al. The Associations of Dietary Intake of Polyunsaturated fatty acids with diabetic retinopathy in well-controlled diabetes. Invest Ophthalmol Vis Sci. 2015, 56(12), 7473–7479.
  7. Alcubierre, N.; Navarrete-Munoz, E.M.; Rubinat, E.; Falguera, M.; Valls, J.; Traveset, A. et al. Association of low oleic acid intake with diabetic retinopathy in type 2 diabetic patients: a casecontrol study. Nutr Metab (Lond). 2016, 13, 40.

Section 8.

We have corrected manucrypt according to the recommendations.

It has been shown that DM patients leading a sedentary life style have a higher risk of diabetic retinopathy as compared to those living actively [1]. Less physically active diabetic patients showed increased blood flow in the retina on exertion [2,3].

Meta-analysis by Umpierre et al. showed that more structured training, following the ADA’s guideline (> 150 min per week), and receiving PA advice alone were correlated with a greater reduction in HbA1c in T2DM patients [4]. 

Meta-analysis by Boniol et al. also suggested a possible mechanism of PA’s impact on DR due to glycemic control improvement [5]. Alteration in the 25-hydroxyvitamin D (25OH-D) level can be another likely mechanism. Substantial evidence has shown that higher PA level contributes to the improved 25OH-D status in people at all ages [6-10]. Keech et al. observed lower blood 25OH-D concentration in relation to a higher risk of macrovascular and microvascular events, including DR [11].

It has been proved that physical exercise is able to modulate oxidative stress [12]. A few experiments have revealed a decrease in oxidative stress in the retinas of DR mice during physical exercise [13-16], and a positive alterations in microglia in rats with streptozotocin-induced DM following treadmill exercise [17].

It has to be remembered, however, that patients suffering from proliferative diabetic retinopathy should avoid high intensity aerobic and resistance exercise to reduce the risk of vitreous hemorrhage or retinal detachment [18,19]. All types of physical exercise that may predispose to elevated systolic blood pressure (Valsalva maneuvers) increase the risk of vitreous hemorrhage [20,21].

  1. Ren, C.; Liu, W.; Li, J.; Cao, Y.; Xu, J.; Lu, P. Physical activity and risk of diabetic retinopathy: A systematic review and meta-analysis. Acta Diabetol. 2019, 56(8), 823–37. doi: 10.1007/s00592-019-01319-4. - DOI – PubMed
  2. Hayashi, N.; Ikemura, T.; Someya, N. Effects of dynamic exercise and its intensity on ocular blood flow in humans. Eur J Appl Physiol. 2011, 111(10), 2601–2606.
  3. Zhang, Y.; San Emeterio Nateras, O.; Peng, Q. et al. Blood flow MRI of the human retina/choroid during rest and isometric exercise. Invest Ophthalmol Vis Sci. 2012, 53(7), 4299–4305.
  4. Umpierre, D.; Ribeiro, P.A.; Kramer, C.K. et al. Physical activity advice only or structured exercise training and association with HbA1c levels in type 2 diabetes: a systematic review and meta-analysis. JAMA. 2011, 305(17), 1790–1799.
  5. Boniol, M.; Dragomir, M. Physical activity and change in fasting glucose and HbA1c: a quantitative meta-analysis of randomized trials. Acta Diabetol. 2017, 54(11), 983–991.
  6. Al-Othman, A.; Al-Musharaf, S.; Al-Daghri, N.M. et al. Effect of physical activity and sun exposure on vitamin D status of Saudi children and adolescents. BMC Pediatr. 2012, 12, 92.
  7. Scott, D.; Blizzard, L.; Fell, J. et al. A prospective study of the associations between 25-hydroxy-vitamin D, sarcopenia progression and physical activity in older adults. Clin Endocrinol (Oxf). 2010, 3(5), 581–587.
  8. Klenk, J.; Rapp, K.; Denkinger, M et al. Objectively measured physical activity and vitamin D status in older people from Germany. J Epidemiol Community Health. 2015, 69(4), 388–392.
  9. Makanae, Y.; Ogasawara, R.; Sato, K. et al. Acute bout of resistance exercise increases vitamin D receptor protein expression in rat skeletal muscle. Exp Physiol. 2015, 100(10), 1168–1176.
  10. Black, L.J.; Burrows, S.A., Jacoby, P. et al. Vitamin D status and predictors of serum 25-hydroxyvitamin D concentrations in Western Australian adolescents. Br J Nutr. 2014, 112(7), 1154–1162.
  11. Keech, A.C., Mitchell, P.; Summanen, P.A. et al. Effect of fenofibrate on the need for laser treatment for diabetic retinopathy (FIELD study): a randomised controlled trial. Lancet. 2007, 370(9600), 1687–1697.
  12. Sallam, N.; Laher, I. Exercise modulates oxidative stress and inflammation in aging and cardiovascular diseases. Oxid Med Cell Longev. 2016, 2016, 7239639.
  13. Kim, C.S.; Park, S.; Chun, Y. et al. Treadmill exercise attenuates retinal oxidative stress in naturally-aged mice: an immunohistochemical study. Int J Mol Sci. 2015, 16(9), 21008–21020.
  14. Kruk, J.; Kubasik-Kladna, K.; Aboul-Enein, H.Y. The role oxidative stress in the pathogenesis of eye diseases: current status and a dual role of physical activity. Mini Rev Med Chem. 2015, 16(3), 241–257.
  15. Allen, R.S.; Hanif, A.M.; Gogniat, M.A. et al. TrkB signalling pathway mediates the protective effects of exercise in the diabetic rat retina. Eur J Neurosci. 2018, 47(10), 1254–1265.
  16. Cui, J.Z.; Wong, M.; Wang, A. et al. Exercise inhibits progression of diabetic retinopathy by reducing inflammatory, oxidative stress, and ER stress gene expression in the retina of db/db mice. Invest Ophthalmol Vis Sci. 2016, 57(12), 5434.
  17. Lu, Y.; Dong, Y.; Tucker, D. et al. Treadmill exercise exerts neuroprotection and regulates microglial polarization and oxidative stress in a streptozotocin-induced rat model of sporadic alzheimer’s disease. J Alzheimers Dis. 2017, 56(4), 1469–1484.
  18. Schneider, S.H.; Khachadurian, A.K.; Amorosa, L.F. et al. Tenyear experience with an exercise-based outpatient life-style modification program in the treatment of diabetes mellitus. Diabetes Care. 1992, 15(11), 1800–1810.
  19. Colberg, S.R. Exercise and diabetes: a clinician’s guide to prescribing physical activity, 1st edn. American Diabetes Association, 2013. Alexandria.
  20. Graham, C.; Lasko-McCarthey, P. Exercise options for persons with diabetic complications. Diabetes Educ. 1990, 16(3), 212–220.
  21. Hamdy, O.; Goodyear, L.J.; Horton, E.S. Diet and exercise in type 2 diabetes mellitus. Endocrinol Metab Clin North Am. 2001, 30(4), 883–907.

Section 8.

We have completed the information.

Research using animal models of diabetes has shown that resistance exercise can cause increased muscle mass [1]. The skeletal muscle is a substantial glucose reservoir in the body and physical exercise is a potent stimulant of glucose uptake partly via the skeletal muscle glucose transporter protein action [2]. That is why resistance training exerting a direct effect on skeletal muscle is likely to play a role in the management of D2T patients [3].

Resistance training is a good alternative to aerobic exercise. This type of training has not been found to cause a greater number of undesirable events as compared to other types of exercise [3]. Pollock et al. reported that there were more complications during jogging and walking than in strength training in the elderly [4].

Compliance with recommendations related to aerobic exercise can be difficult. Aerobic training should be conducted almost every day or even all the week to be effective. However, progressive resistance exercise can be effective if performed only three times a week [3].

Moreover, some coexisting diseases in diabetic patients (cardiovascular and peripheral vascular disorders, neuropathy and motor impairment – foot ulceration, claudication and a risk of falling down) may hinder aerobic performance. In all these situations, resistance training is not only a real alternative but can also be more viable than aerobic exercise. In patients who can safely take aerobic exercise, a combined training program is more effective than the progressive resistance training program alone [3]. Sigal et al. reported that the levels of glycosylated hemoglobin decreased markedly in the combined training group as compared to the only aerobic or resistance training groups [5].

Eight weeks with two or three 45 min. sessions of progressive training are sufficient to improve glycemic control [3].

A reduction in glycosylated hemoglobin by 1% is associated with a 37% decrease in the risk of microvascular complications and a 21% decrease in the risk of diabetes-related death [6].

Snowling and Hopkins observed a 0.5% reduction in glycosylated hemoglobin after progressive resistance training [7].

  1. Farrell, P.; Fedele, M.; Hernandez, J.; Fluckey, J.; Miller, J.; Lang, C. et al. Hypertrophy of skeletal muscle in diabetic rats in response to chronic resistance exercise. Journal of Applied Physiology. 1999, 87, 1075–1082.
  2. Schuller, G.; Linke, A. Diabetes, exercise. In Goldstein, B.; Muller-Wieland, D. (Eds) Type 2 Diabetes: Principles, Practice. New York: Informa Healthcare, Ch 6. 2008
  3. Irvine, C.; Taylor, N.C. Progressive resistance exercise improves glycaemic control in people with type 2 diabetes mellitus: a systematic review. Australian Journal of Physiotherapy. 2009, 55, 237-246.
  4. Pollock, M.; Carroll, J.; Graves, J.; Leggett, S.; Braith, R.; Limacher, M. et al. Injuries, adherence to walk/jog, resistance training programs in the elderly. Medicine, Science in Sports, Exercise. 1991, 23, 1194–1200.
  5. Sigal, R.; Kenny, G.; Boule, N.; Wells, G.; Prud’homme, D.; Fortier, M. et al. Effects of aerobic training, resistance training, or both on glycemic control in type 2 diabetes: A randomized trial. Annals of Internal Medicine. 2007, 147, 357–369.
  6. Stratton, I.; Adler, A.; Neil, H.; Matthews, D.; Manley, S.; Cull, C. et al. Association of glycemia with macrovascular, microvascular complications of type 2 diabetes (UKPDS35): Prospective observational study. BMJ. 2000, 321, 405–412.
  7. Snowling, N.; Hopkins, W. Effects of different modes of exercise training on glucose control, risk factors for complications in type 2 diabetic patients: a meta-analysis. Diabetes Care. 2006, 29, 2518–2527.

Section 11.

We added more information.

Chlorogenic acid (CGA) is a polyphenol found in a variety of products, such as coffee, grains, potatoes and apples. It is the ester of caffeic and quinic acids, showing antibacterial, antiinflammatory, antioxidant and anineoplastic effects [1-3]. CGA can delay glucose absorption in the small intestine and decrease glucose production in the liver [4,5]. It is believed that CGA stimulates secretion of glucagon-like peptide 1 known to exert a beneficial effect on the response to glucose in pancreatic beta cells [3]. In the liver, CGA inhibits glucose-6-phospatase, thus decreasing hepatic glucose production [5,6]. CGA reduces hyperpermeability of retinal vessels in diabetic rats. The diabetic rats with higher level of VEGF and down-regulation of occludin, claudin-5 and ZO-1 showed breakdown of the blood-retina barrier and intensified vascular leakage. CGA managed to maintain occludin expression and reduced the levels of VEGF, which decreased the BRB breakdown and inhibited vascular leakage. Thus, CGA may prevent BRB breakdown in retinopathy [7].

  1. Dos Santos, M.D.; Almeida, M.C.; Lopes, N.P.; de Souza, G.E. Evaluation of the anti-inflammatory, analgesic and antipyretic activities of the natural polyphenol chlorogenic acid. Biol Pharm Bull. 2006, 29, 2236-2240.
  2. Puupponen-Pimiä, R.; Nohynek, L.; Meier, C.; Kähkönen, M.; Heinonen, M.; Hopia, A, Oksman-Caldentey, K.M. Antimicrobial properties of phenolic compounds from berries. J Appl Microbiol. 2001, 90, 494-507.
  3. Ma, C.M.; Kully, M.; Khan, J.K.; Hattori, M.; Daneshtalab, M. Synthesis of chlorogenic acid derivatives with promising antifungal activity. Bioorg Med Chem. 2007, 15, 6830-6833.
  4. McCarty, M.F. A chlorogenic acid-induced increase in GLP-1 production may mediate the impact of heavy coffee consumption on diabetes risk. Med Hypotheses. 2005, 64, 848-853.
  5. Arion, W.J.; Canfield, W.K.; Ramos, F.C. Schindler, P.W.; Burger, H.J.; Hemmerle, H.; Schubert, G.; Below, P.; Herling, AW. Chlorogenic acid and hydroxynitrobenzaldehyde: new inhibitors of hepatic glucose 6-phosphatase. Arch Biochem Biophys. 1997, 339, 315-322.
  6. Herling, A.W.; Burger, H.; Schubert, G.; Hemmerle, H.; Schaefer, H.; Kramer, W. Alterations of carbohydrate and lipid intermediary metabolism during inhibition of glucose-6-phosphatase in rats. Eur J Pharmacol. 1999, 386, 75-82.
  7. Shin, J.Y.; Sohn, J.; Hyung, K. ParkChlorogenic Acid Decreases Retinal Vascular Hyperpermeability in Diabetic Rat Model. Korean Med Sci .2013, 28, 608-613.

Calcium dobesilate (CaD) is considered to be an angioprotective drug. For many years CaD was used in diabetic retinopathy [4,5]. Researchers investigating its efficacy are divided. A few clinical studies showed delayed progression of diabetic retinopathy after long-term oral treatment with CaD. It was found to inhibit changes in tight junction proteins, ICAM-1 and leukocyte adhesion to retinal vessels which are known to lie at the base of growing permeability of the blood-retina barrier. These results were correlated with the inhibition of oxidative / nitrosative stress and the activity of  p38 MAPK and NF-κB [1,2]. A positive effect of CaD (2000 mg/day for 2 years) was found in patients suffering from early diabetic retinopathy [6]. Other studies failed to show a positive effect of CaD on the retina of diabetic patients [8,9].  In a recent study (CALDIRET) (30), in which patients with mild to moderate non-proliferative diabetic retinopathy were followed-up for 5 years, CaD did not reduce diabetic macular edema [7].

  1. Leal, E.C.; Martins, J.; Voabil, P.; Liberal, J.; Chiavaroli, C.; Bauer, J.; Cunha-Vaz, J.; Ambrósio, A.F. Calcium Dobesilate Inhibits the Alterations in Tight Junction Proteins and Leukocyte Adhesion to Retinal Endothelial Cells Induced by Diabetes 2010, 59(10), 2637–2645. Published online 2010 Jul 13. doi: 10.2337/db09-1421
  2. Adamis, A.P.; Berman, A.J. Immunological mechanisms in the pathogenesis of diabetic retinopathy. Semin Immunopathol. 2008, 30, 65–84. [PubMed] [Google Scholar]
  3. Kern, T.S. Contributions of inflammatory processes to the development of the early stages of diabetic retinopathy. Exp Diabetes Res. 2007, 2007, 95103. [PMC free article] [PubMed] [Google Scholar]
  4. Berthet, P.; Farine, J.C.; Barras, J.P. Calcium dobesilate: pharmacological profile related to its use in diabetic retinopathy. Int J Clin Pract. 1999, 53, 631–636. [PubMed] [Google Scholar]
  5. Tejerina, T.; Ruiz, E. Calcium dobesilate: pharmacology and future approaches. Gen Pharmacol. 1998, 31, 357–360.
  6. Ribeiro, M.L.; Seres, A.I.; Carneiro, A.M.; Stur, M.; Zourdani, A.; Caillon, P.; Cunha-Vaz, J.G. DX-Retinopathy Study Group Effect of calcium dobesilate on progression of early diabetic retinopathy: a randomised double-blind study. Graefes Arch Clin Exp Ophthalmol. 2006, 244, 1591–1600.
  7. Haritoglou, C.; Gerss, J.; Sauerland, C.; Kampik, A.; Ulbig, M.W. CALDIRET study group Effect of calcium dobesilate on occurrence of diabetic macular oedema (CALDIRET study): randomised, double-blind, placebo-controlled, multicentre trial. Lancet. 2009, 373, 1364–1371.
  8. Liu, J.; Li, S.; Sun, D. Calcium Dobesilate and Micro-vascular diseases. Life Sci. 2019, Mar 15, 221, 348-353. doi: 10.1016/j.lfs.2019.02.023. Epub 2019 Feb 12.
  9. Solà-Adell, C.; Bogdanov, P.Hernández, C.; Sampedro, J.; Valeri, M.; Garcia-Ramirez, M.; Pasquali, C.; Simó, R. Calcium Dobesilate Prevents Neurodegeneration and Vascular Leakage in Experimental Diabetes. Curr Eye Res. 2017, 42(9), 1273-1286. doi: 10.1080/02713683.2017.1302591. Epub 2017 Jun 2

It has been found that n3-PUFA can reduce the number of retinal acellular capillaries in diabetes and inflammatory markers in the retina of diabetic animals [1,2]. Diabetic patients taking n-3 PUFA supplements for 18 months had lower risk of retinopathy [3].

The effect of vitamin C intake on DR is not certain [4-6]. The use of vitamin C supplements may suggest that this vitamin is involved in retinopathy prevention [4].

Millen et al. [5], conducting a prospective study of the ARIC participants,  failed to observe an effect of vitamin C and E intake from food alone or a combination of food and supplements on the risk of retinopathy. However, they found an interaction between race and vitamin E; high consumption of vitamin E from food alone, or food combined with supplements, was associated with higher prevalence of retinopathy in Caucasians, but not in African Americans [5]. Diminished likelihood of retinopathy was noted in those taking (C 3 years) vitamin C, E, and multivitamin supplements in comparison with non-users [5]. However, in a cross-sectional analysis of NHANES III participants, Millen et al. found no relationship between taking vitamin C or E supplements for a long time (C 5 vs.\1 year) and the presence of retinopathy [7].

The intake of vitamin D and fish oil supplements was not associated with the risk of retinopathy in the ARIC [8].

Lutein intake was not significantly correlated with retinopathy [9]. In one study, the serum level of lutein was found to be decreased in patients with nonproliferative diabetic retinopathy as compared to diabetics without retinopathy [10].

Another study suggested a positive impact of high plasma level of lutein/zeaxanthin and lycopene on the risk of diabetic retinopathy [11]. However, increased consumption of b-carotene was found to be related to lower risk and was not associated with retinopathy [4,6].

  1. Tikhonenko, M.; Lydic, T.A.; Opreanu, M.; Li, C.S.; Bozack, S.; McSorley, K.M. et al. N-3 polyunsaturated Fatty acids prevent diabetic retinopathy by inhibition of retinal vascular damage and enhanced endothelial progenitor cell reparative function. PLoS ONE. 2013, 8(1), e55177.
  2. Shen, J.H.; Ma, Q.; Shen, S.R.; Xu, G.T.; Das, U.N. Effect of alphalinolenic acid on streptozotocin-induced diabetic retinopathy indices in vivo. Arch Med Res. 2013, 44(7), 514–520.
  3. Roig-Revert, M.J.; Lleo-Perez, A.; Zanon-Moreno, V.; Vivar-Llopis, B.; Marin-Montiel, J.; Dolz-Marco, R. et al. Enhanced oxidative stress and other potential biomarkers for retinopathy in type 2 diabetics: beneficial effects of the nutraceutic supplements. Biomed Res Int. 2015, 2015, 408180.
  4. Tanaka, S.; Yoshimura, Y.; Kawasaki, R.; Kamada, C.; Tanaka, S.; Horikawa, C. et al. Fruit intake and incident diabetic retinopathy with type 2 diabetes. 2013, 24(2), 204–201.
  5. Millen, A.E.; Klein, R.; Folsom, A.R.; Stevens, J.; Palta, M.; Mares, J.A. Relation between intake of vitamins C and E and risk of diabetic retinopathy in the Atherosclerosis Risk in Communities Study. Am J Clin Nutr. 2004, 79(5), 865–873.
  6. Mayer-Davis, E.J.; Bell, R.A.; Reboussin, B.A.; Rushing, J.; Marshall, J.A.; Hamman, R.F. Antioxidant nutrient intake and diabetic retinopathy: the San Luis Valley Diabetes Study. Ophthalmology. 1998, 105(12), 2264–2270.
  7. Millen, A.E.; Gruber, M.; Klein, R.; Klein, B.E.; Palta, M.; Mares, J.A. Relations of serum ascorbic acid and alpha-tocopherol to diabetic retinopathy in the Third National Health and Nutrition Examination Survey. Am J Epidemiol. 2003, 158(3), 225–233.
  8. Millen, A.E.; Sahli, M.W.; Nie, J.; LaMonte, M.J.; Lutsey, P.L.; Klein, B.E. et al. Adequate vitamin D status is associated with the reduced odds of prevalent diabetic retinopathy in African Americans and Caucasians. Cardiovasc Diabetol. 2016, 15(1), 128.
  9. Sahli, M.W.; Mares, J.A.; Meyers, K.J.; Klein, R.; Brady, W.E.; Klein, B.E. et al. Dietary intake of lutein and diabetic retinopathy in the atherosclerosis risk in Communities Study (ARIC). Ophthalmic Epidemiol. 2016, 23(2), 99–108.
  10. Koushan, K.; Rusovici, R. Li, W.; Ferguson, L.R.; Chalam, K.V. The role of lutein in eye-related disease. 2013, 5(5), 1823–1839.
  11. Brazionis, L.; Rowley, K.; Itsiopoulos, C.; O’Dea, K. Plasma carotenoids and diabetic retinopathy. Br J Nutr. 2009, 101(2), 270–277.

    Response to Rewiever 3

    Thank you very much for your important comments.

    Section 7.

    We have completed the information concerning the effect of coffee and tea intake on the risk of diabetic retinopathy.

    It has been shown that tea acts as a potent neuroprotector in the retina [5], preventing formation of acellular capillary vessels and pericyte ghosts in diabetic rats [1]. Green tea can protect diabetic retinal neurons and regulate the subretinal environment by decreasing  ROS generation due to increased expression of glutamate transporter, restored intercellular connections and glutamine/glutamate circulation [2]. Moreover, a low dose of green tea  is likely to improve antioxidant defense, decrease inflammatory markers and prevent thickening of the basement membrane of the retina [3]. Black tea, which lowers blood sugar and thus inhibits pathological biochemical indices, is able to delay diabetic cataract development  [4].

    As found in a Chinese case–control study, involving sex and age-matched controls suffering from diabetes but not diabetic retinopathy, regular consumption (every week for not less than 1 year) of Chinese green tea was associated with reduced probability of retinopathy in women, but not in men [6].

    1. Mustata, G.T.; Rosca, M.; Biemel, K.M.; Reihl, O.; Smith, M.A.; Viswanathan, A.; Strauch, C.; Du, Y.; Tang, J.; Kern, T.S. et al. Paradoxical effects of green tea (Camellia Sinensis) and antioxidant vitamins in diabetic rats: Improved retinopathy and renal mitochondrial defects but deterioration of collagen matrix glycoxidation and cross-linking. Diabetes. 2005, 54, 517–526. doi: 10.2337/diabetes.54.2.517. [PubMed] [CrossRef] [Google Scholar]
    2. Silva, K.C.; Rosales, M.A.B.; Hamassaki, D.E.; Saito, K.C.; Faria, A.M.; Ribeiro, P.A.O.; de Faria, J.B.L.; de Faria, J.M.L. Green tea is neuroprotective in diabetic retinopathy. Ophthalmol. Vis. Sci. 2013, 54, 1325–1336. doi: 10.1167/iovs.12-10647. [PubMed] [CrossRef] [Google Scholar]
    3. Kumar, B.; Gupta, S.K.; Nag, T.C.; Srivastava, S.; Saxena R. Green tea prevents hyperglycemia-induced retinal oxidative stress and inflammation in streptozotocin-induced diabetic rats. Ophthalmic Res. 2012, 47, 103–108. doi: 10.1159/000330051. [PubMed] [CrossRef] [Google Scholar]
    4. Vinson, J.A.; Zhang, J. Black and green teas equally inhibit diabetic cataracts in a streptozotocin-induced rat model of diabetes. Agric. Food Chem. 2005, 53, 3710–3713. doi: 10.1021/jf048052l. [PubMed] [CrossRef] [Google Scholar]
    5. Jin-Ming, Meng.; Shi-Yu, Cao.; Xin-Lin Wei, Ren-You Gan, Yuan-Feng Wang, Shu-Xian Cai, Xiao-Yu Xu, Pang-Zhen Zhang, Hua-Bin Li. Effects and Mechanisms of Tea for the Prevention and Management of Diabetes Mellitus and Diabetic Complications: An Updated Review. Antioxidants (Basel) 2019 Jun; 8(6): 170. Published online 2019 Jun 10. doi: 10.3390/antiox8060170.
    6. Ma, Q.; Chen, D.; Sun, H.P.; Yan, N.; Xu, Y.; Pan, C.W. Regular Chinese green tea consumption is protective for diabetic retinopathy: a Clinic-Based Case-Control Study. J Diabetes Res. 2015, 2015, 231570.

    Coffee is one of the most consumed beverages worldwide. The main commercial coffee blends are Arabica (Coffea arabica L.) and Robusta (Coffea canephora Pierre ex Froehner) [1]. Its main component is caffeine which belongs to antioxidants and its long-term intake in large quantities reduces oxidative stress [2]. Other components of coffee beans include carbohydrates, proteins, fats, alkaloids, diterpenes, free amino acids, melanoides and minerals, both macro- and microelements [3]. Coffee contains microelements that show antioxidant effects, such as manganese, zinc, copper and iron [1]. It can also be a source of fluorine, chromium and cobalt [1].

    A Norwegian cohort study revealed an approximately 35% reduced risk of T2D associated with high (in comparison to low) consumption of brewed coffee and other types of coffee [2]. Likewise, the results of a Finnish cohort study revealed that both the intake of boiled coffee and consumption of drip coffee were inversely correlated with T2D [3].

    A number of coffee components have been found to show anti-inflammatory effects (i.e. caffeine, CGA, cafestol, kahweol, trigonelline, caffeic and ferulic acids). Caffeine and CGA exert an effect on insulin and glucose homeostasis by modulating adenosine receptor signaling, suppressing intestinal glucose absorption (by enhanced generation of gastric inhibitory peptide-1 [GIP-1] and glucagon-like-peptide-1 [GLP-1] or glucose-6-phosphate translocase 1 inhibition), reducing glucose output in the liver (by glucose-6-phosphatase suppression), and by improving secretion of pancreatic islet insulin or peripheral insulin sensitivity and glucose uptake (by glucose transporter type 4 [GLUT4] stimulation and modulation of intracellular signaling pathway activation - Akt, AMPK, MAPK) [4].

    Regular coffee intake can lower the level of proinflammatory biomarkers, such as  L-1β, IL-6, TNF-α, C-reactive protein, monocyte chemotactic protein 1, vascular cell adhesion molecule 1, C-peptides, endothelial-leukocyte adhesion molecule 1, and interleukin 18 [IL-18]) in healthy, obese and T2D-affected individuals. However, it may increase the levels of anti-inflammatory adiponectin, interleukin 4 and interleukin 10  [5,6].

    1. Hjellvik, V.; Tverdal, A.; Strom, H. Boiled coffee intake and subsequent risk for type 2 diabetes. 2011, 22, 418–421.
    2. Olechno, E.; Pus´cion-Jakubik, A.; Socha, K.; Zujko, M.E. Coffee Infusions: Can They Be a Source of Microelements with Antioxidant Properties? Antioxidants (Basel). 2021, Oct 27, 10(11), doi: 10.3390/antiox10111709
    3. Tuomilehto, J.; Hu, G.; Bidel, Set al. , Coffee consumption and risk of type 2 diabetes mellitus among middle-aged Finnish men and women. JAMA. 2004, 291, 1213–1219.
    4. Carlström, M.; Larsson, S.C. Coffee consumption and reduced risk of developing type 2 diabetes: A systematic review with meta-analysis. Rev. 2018, 76, 395–417. [CrossRef] [PubMed]
    5. Natella, F.; Scaccini, C. Role of coffee in modulation of diabetes risk. Nutr Rev. 2012, 70, 207–217.
    6. Akash, M.S.; Rehman, K.; Chen, S. Effects of coffee on type 2 diabetes mellitus. 2014, 30, 755–763.

    Section 9.

    We have completed the information.

    Diaz-Lopez et al. studied the Mediterranean diet enhanced with extra virgin olive oil or nuts,  and compared it with a low-fat diet in more than 3600 participants in a prospective 6-year study. The Mediterranean diet enhanced with olive oil was associated with over 40% reduced risk of retinopathy [1].  The Mediterranean diet enriched with nuts was correlated with a 37% reduced, though statistically insignificant decrease in the risk of retinopathy [1].

    A Japanese cohort study of T2D individuals revealed that high fruit intake (C 173.2 g per day, i.e. an apple or two bananas), was associated with over a 50% reduction in the risk of incident retinopathy, as compared to patients who consumed less than 53.2 g of fruit daily [2].

    Milk intake has also been assessed; the consumption of whole milk and skim/low fat milk was not found to be associated with retinopathy in more than 1350 patients with type 2 diabetes [3].

    Fish may inhibit the development of retinopathy. The intake of oily fish at least twice a week (as compared to more rare consumption), was associated with a nearly 60% reduction in the risk of retinopathy [4]. Another study reported that 85–141 g of dark fish, (salmon, mackerel, swordfish, sardines, bluefish) consumed weekly versus never was related to almost 70% reduced likelihood of retinopathy [3]. However, 85–141 g of ‘‘other fish’’ (cod, perch, catfish), eaten every week was not associated with retinopathy [3].

    The intake of vitamin D and fish oil supplements, versus the non-use, was not related to the risk of retinopathy in the ARIC [3]. In an RCT, diabetics who were randomly given n-3 PUFA supplements for 18 months had a lower risk of retinopathy [5].

    Studies on the effect of fat consumption on the development of diabetic retinopathy are divided. No direct correlation was found between the intake of fat, trans fat, total saturated fatty acid (SFA) and diabetic retinopathy [6,7].

    Sasaki et al. failed to find the  impact of MUFA on retinopathy [6]. However, Alcubierre et al.  reported an inverse correlation of MUFA and oleic acid with odds of retinopathy [7].

    Some evidence indicates that the intake of PUFA may contribute to the prevention of  retinopathy [6]. However, one study found no such correlations [7].

    Considering fiber intake, two studies observed no association with retinopathy, and one Indian study found an inverse correlation [2, 7] .

    1. Diaz-Lopez A, Babio N, Martinez-Gonzalez MA, Corella D, Amor AJ, Fito M, et al. Mediterranean diet, retinopathy, nephropathy, and microvascular diabetes complications: a post hoc analysis of a randomized trial. Diabetes Care. 2015, 38(11), 2134–2141.
    2. Tanaka, S.; Yoshimura, Y.; Kawasaki, R.; Kamada, C.; Tanaka, S.; Horikawa, C. et al. Fruit intake and incident diabetic retinopathy with type 2 diabetes. 2013, 24(2), 204–211.
    3. Millen, A.E.; Sahli, M.W.; Nie, J.; LaMonte, M.J.; Lutsey, P.L.; Klein, B.E. et al. Adequate vitamin D status is associated with the reduced odds of prevalent diabetic retinopathy in African Americans and Caucasians. Cardiovasc. 2016, 15(1), 128.
    4. Sala-Vila, A.; Diaz-Lopez, A.; Valls-Pedret, C.; Cofan, M.; GarciaLayana, A.; Lamuela-Raventos, R.M. et al. Dietary Marine omega-3 fatty acids and incident sight-threatening retinopathy in middleaged and older individuals with type 2 diabetes: prospective investigation from the PREDIMED trial. JAMA Ophthalmol. 2016, 134(10), 1142–1149.
    5. Roig-Revert, M.J.; Lleo-Perez, A.; Zanon-Moreno, V.; Vivar-Llopis, B.; Marin-Montiel, J.; Dolz-Marco, R. et al. Enhanced oxidative stress and other potential biomarkers for retinopathy in type 2 diabetics: beneficial effects of the nutraceutic supplements. Biomed Res Int. 2015, 2015, 408180.
    6. Sasaki, M.; Kawasaki, R.; Rogers, S.; Man, R.E.; Itakura, K.; Xie, J. et al. The Associations of Dietary Intake of Polyunsaturated fatty acids with diabetic retinopathy in well-controlled diabetes. Invest Ophthalmol Vis Sci. 2015, 56(12), 7473–7479.
    7. Alcubierre, N.; Navarrete-Munoz, E.M.; Rubinat, E.; Falguera, M.; Valls, J.; Traveset, A. et al. Association of low oleic acid intake with diabetic retinopathy in type 2 diabetic patients: a casecontrol study. Nutr Metab (Lond). 2016, 13, 40.

    Section 8.

    We have corrected manucrypt according to the recommendations.

    It has been shown that DM patients leading a sedentary life style have a higher risk of diabetic retinopathy as compared to those living actively [1]. Less physically active diabetic patients showed increased blood flow in the retina on exertion [2,3].

    Meta-analysis by Umpierre et al. showed that more structured training, following the ADA’s guideline (> 150 min per week), and receiving PA advice alone were correlated with a greater reduction in HbA1c in T2DM patients [4]. 

    Meta-analysis by Boniol et al. also suggested a possible mechanism of PA’s impact on DR due to glycemic control improvement [5]. Alteration in the 25-hydroxyvitamin D (25OH-D) level can be another likely mechanism. Substantial evidence has shown that higher PA level contributes to the improved 25OH-D status in people at all ages [6-10]. Keech et al. observed lower blood 25OH-D concentration in relation to a higher risk of macrovascular and microvascular events, including DR [11].

    It has been proved that physical exercise is able to modulate oxidative stress [12]. A few experiments have revealed a decrease in oxidative stress in the retinas of DR mice during physical exercise [13-16], and a positive alterations in microglia in rats with streptozotocin-induced DM following treadmill exercise [17].

    It has to be remembered, however, that patients suffering from proliferative diabetic retinopathy should avoid high intensity aerobic and resistance exercise to reduce the risk of vitreous hemorrhage or retinal detachment [18,19]. All types of physical exercise that may predispose to elevated systolic blood pressure (Valsalva maneuvers) increase the risk of vitreous hemorrhage [20,21].

    1. Ren, C.; Liu, W.; Li, J.; Cao, Y.; Xu, J.; Lu, P. Physical activity and risk of diabetic retinopathy: A systematic review and meta-analysis. Acta Diabetol. 2019, 56(8), 823–37. doi: 10.1007/s00592-019-01319-4. - DOI – PubMed
    2. Hayashi, N.; Ikemura, T.; Someya, N. Effects of dynamic exercise and its intensity on ocular blood flow in humans. Eur J Appl Physiol. 2011, 111(10), 2601–2606.
    3. Zhang, Y.; San Emeterio Nateras, O.; Peng, Q. et al. Blood flow MRI of the human retina/choroid during rest and isometric exercise. Invest Ophthalmol Vis Sci. 2012, 53(7), 4299–4305.
    4. Umpierre, D.; Ribeiro, P.A.; Kramer, C.K. et al. Physical activity advice only or structured exercise training and association with HbA1c levels in type 2 diabetes: a systematic review and meta-analysis. JAMA. 2011, 305(17), 1790–1799.
    5. Boniol, M.; Dragomir, M. Physical activity and change in fasting glucose and HbA1c: a quantitative meta-analysis of randomized trials. Acta Diabetol. 2017, 54(11), 983–991.
    6. Al-Othman, A.; Al-Musharaf, S.; Al-Daghri, N.M. et al. Effect of physical activity and sun exposure on vitamin D status of Saudi children and adolescents. BMC Pediatr. 2012, 12, 92.
    7. Scott, D.; Blizzard, L.; Fell, J. et al. A prospective study of the associations between 25-hydroxy-vitamin D, sarcopenia progression and physical activity in older adults. Clin Endocrinol (Oxf). 2010, 3(5), 581–587.
    8. Klenk, J.; Rapp, K.; Denkinger, M et al. Objectively measured physical activity and vitamin D status in older people from Germany. J Epidemiol Community Health. 2015, 69(4), 388–392.
    9. Makanae, Y.; Ogasawara, R.; Sato, K. et al. Acute bout of resistance exercise increases vitamin D receptor protein expression in rat skeletal muscle. Exp Physiol. 2015, 100(10), 1168–1176.
    10. Black, L.J.; Burrows, S.A., Jacoby, P. et al. Vitamin D status and predictors of serum 25-hydroxyvitamin D concentrations in Western Australian adolescents. Br J Nutr. 2014, 112(7), 1154–1162.
    11. Keech, A.C., Mitchell, P.; Summanen, P.A. et al. Effect of fenofibrate on the need for laser treatment for diabetic retinopathy (FIELD study): a randomised controlled trial. Lancet. 2007, 370(9600), 1687–1697.
    12. Sallam, N.; Laher, I. Exercise modulates oxidative stress and inflammation in aging and cardiovascular diseases. Oxid Med Cell Longev. 2016, 2016, 7239639.
    13. Kim, C.S.; Park, S.; Chun, Y. et al. Treadmill exercise attenuates retinal oxidative stress in naturally-aged mice: an immunohistochemical study. Int J Mol Sci. 2015, 16(9), 21008–21020.
    14. Kruk, J.; Kubasik-Kladna, K.; Aboul-Enein, H.Y. The role oxidative stress in the pathogenesis of eye diseases: current status and a dual role of physical activity. Mini Rev Med Chem. 2015, 16(3), 241–257.
    15. Allen, R.S.; Hanif, A.M.; Gogniat, M.A. et al. TrkB signalling pathway mediates the protective effects of exercise in the diabetic rat retina. Eur J Neurosci. 2018, 47(10), 1254–1265.
    16. Cui, J.Z.; Wong, M.; Wang, A. et al. Exercise inhibits progression of diabetic retinopathy by reducing inflammatory, oxidative stress, and ER stress gene expression in the retina of db/db mice. Invest Ophthalmol Vis Sci. 2016, 57(12), 5434.
    17. Lu, Y.; Dong, Y.; Tucker, D. et al. Treadmill exercise exerts neuroprotection and regulates microglial polarization and oxidative stress in a streptozotocin-induced rat model of sporadic alzheimer’s disease. J Alzheimers Dis. 2017, 56(4), 1469–1484.
    18. Schneider, S.H.; Khachadurian, A.K.; Amorosa, L.F. et al. Tenyear experience with an exercise-based outpatient life-style modification program in the treatment of diabetes mellitus. Diabetes Care. 1992, 15(11), 1800–1810.
    19. Colberg, S.R. Exercise and diabetes: a clinician’s guide to prescribing physical activity, 1st edn. American Diabetes Association, 2013. Alexandria.
    20. Graham, C.; Lasko-McCarthey, P. Exercise options for persons with diabetic complications. Diabetes Educ. 1990, 16(3), 212–220.
    21. Hamdy, O.; Goodyear, L.J.; Horton, E.S. Diet and exercise in type 2 diabetes mellitus. Endocrinol Metab Clin North Am. 2001, 30(4), 883–907.

    Section 8.

    We have completed the information.

    Research using animal models of diabetes has shown that resistance exercise can cause increased muscle mass [1]. The skeletal muscle is a substantial glucose reservoir in the body and physical exercise is a potent stimulant of glucose uptake partly via the skeletal muscle glucose transporter protein action [2]. That is why resistance training exerting a direct effect on skeletal muscle is likely to play a role in the management of D2T patients [3].

    Resistance training is a good alternative to aerobic exercise. This type of training has not been found to cause a greater number of undesirable events as compared to other types of exercise [3]. Pollock et al. reported that there were more complications during jogging and walking than in strength training in the elderly [4].

    Compliance with recommendations related to aerobic exercise can be difficult. Aerobic training should be conducted almost every day or even all the week to be effective. However, progressive resistance exercise can be effective if performed only three times a week [3].

    Moreover, some coexisting diseases in diabetic patients (cardiovascular and peripheral vascular disorders, neuropathy and motor impairment – foot ulceration, claudication and a risk of falling down) may hinder aerobic performance. In all these situations, resistance training is not only a real alternative but can also be more viable than aerobic exercise. In patients who can safely take aerobic exercise, a combined training program is more effective than the progressive resistance training program alone [3]. Sigal et al. reported that the levels of glycosylated hemoglobin decreased markedly in the combined training group as compared to the only aerobic or resistance training groups [5].

    Eight weeks with two or three 45 min. sessions of progressive training are sufficient to improve glycemic control [3].

    A reduction in glycosylated hemoglobin by 1% is associated with a 37% decrease in the risk of microvascular complications and a 21% decrease in the risk of diabetes-related death [6].

    Snowling and Hopkins observed a 0.5% reduction in glycosylated hemoglobin after progressive resistance training [7].

    1. Farrell, P.; Fedele, M.; Hernandez, J.; Fluckey, J.; Miller, J.; Lang, C. et al. Hypertrophy of skeletal muscle in diabetic rats in response to chronic resistance exercise. Journal of Applied Physiology. 1999, 87, 1075–1082.
    2. Schuller, G.; Linke, A. Diabetes, exercise. In Goldstein, B.; Muller-Wieland, D. (Eds) Type 2 Diabetes: Principles, Practice. New York: Informa Healthcare, Ch 6. 2008
    3. Irvine, C.; Taylor, N.C. Progressive resistance exercise improves glycaemic control in people with type 2 diabetes mellitus: a systematic review. Australian Journal of Physiotherapy. 2009, 55, 237-246.
    4. Pollock, M.; Carroll, J.; Graves, J.; Leggett, S.; Braith, R.; Limacher, M. et al. Injuries, adherence to walk/jog, resistance training programs in the elderly. Medicine, Science in Sports, Exercise. 1991, 23, 1194–1200.
    5. Sigal, R.; Kenny, G.; Boule, N.; Wells, G.; Prud’homme, D.; Fortier, M. et al. Effects of aerobic training, resistance training, or both on glycemic control in type 2 diabetes: A randomized trial. Annals of Internal Medicine. 2007, 147, 357–369.
    6. Stratton, I.; Adler, A.; Neil, H.; Matthews, D.; Manley, S.; Cull, C. et al. Association of glycemia with macrovascular, microvascular complications of type 2 diabetes (UKPDS35): Prospective observational study. BMJ. 2000, 321, 405–412.
    7. Snowling, N.; Hopkins, W. Effects of different modes of exercise training on glucose control, risk factors for complications in type 2 diabetic patients: a meta-analysis. Diabetes Care. 2006, 29, 2518–2527.

    Section 11.

    Dodaliśmy zalecane informacje.

    Chlorogenic acid (CGA) is a polyphenol found in a variety of products, such as coffee, grains, potatoes and apples. It is the ester of caffeic and quinic acids, showing antibacterial, antiinflammatory, antioxidant and anineoplastic effects [1-3]. CGA can delay glucose absorption in the small intestine and decrease glucose production in the liver [4,5]. It is believed that CGA stimulates secretion of glucagon-like peptide 1 known to exert a beneficial effect on the response to glucose in pancreatic beta cells [3]. In the liver, CGA inhibits glucose-6-phospatase, thus decreasing hepatic glucose production [5,6]. CGA reduces hyperpermeability of retinal vessels in diabetic rats. The diabetic rats with higher level of VEGF and down-regulation of occludin, claudin-5 and ZO-1 showed breakdown of the blood-retina barrier and intensified vascular leakage. CGA managed to maintain occludin expression and reduced the levels of VEGF, which decreased the BRB breakdown and inhibited vascular leakage. Thus, CGA may prevent BRB breakdown in retinopathy [7].

    1. Dos Santos, M.D.; Almeida, M.C.; Lopes, N.P.; de Souza, G.E. Evaluation of the anti-inflammatory, analgesic and antipyretic activities of the natural polyphenol chlorogenic acid. Biol Pharm Bull. 2006, 29, 2236-2240.
    2. Puupponen-Pimiä, R.; Nohynek, L.; Meier, C.; Kähkönen, M.; Heinonen, M.; Hopia, A, Oksman-Caldentey, K.M. Antimicrobial properties of phenolic compounds from berries. J Appl Microbiol. 2001, 90, 494-507.
    3. Ma, C.M.; Kully, M.; Khan, J.K.; Hattori, M.; Daneshtalab, M. Synthesis of chlorogenic acid derivatives with promising antifungal activity. Bioorg Med Chem. 2007, 15, 6830-6833.
    4. McCarty, M.F. A chlorogenic acid-induced increase in GLP-1 production may mediate the impact of heavy coffee consumption on diabetes risk. Med Hypotheses. 2005, 64, 848-853.
    5. Arion, W.J.; Canfield, W.K.; Ramos, F.C. Schindler, P.W.; Burger, H.J.; Hemmerle, H.; Schubert, G.; Below, P.; Herling, AW. Chlorogenic acid and hydroxynitrobenzaldehyde: new inhibitors of hepatic glucose 6-phosphatase. Arch Biochem Biophys. 1997, 339, 315-322.
    6. Herling, A.W.; Burger, H.; Schubert, G.; Hemmerle, H.; Schaefer, H.; Kramer, W. Alterations of carbohydrate and lipid intermediary metabolism during inhibition of glucose-6-phosphatase in rats. Eur J Pharmacol. 1999, 386, 75-82.
    7. Shin, J.Y.; Sohn, J.; Hyung, K. ParkChlorogenic Acid Decreases Retinal Vascular Hyperpermeability in Diabetic Rat Model. Korean Med Sci .2013, 28, 608-613.

    Calcium dobesilate (CaD) is considered to be an angioprotective drug. For many years CaD was used in diabetic retinopathy [4,5]. Researchers investigating its efficacy are divided. A few clinical studies showed delayed progression of diabetic retinopathy after long-term oral treatment with CaD. It was found to inhibit changes in tight junction proteins, ICAM-1 and leukocyte adhesion to retinal vessels which are known to lie at the base of growing permeability of the blood-retina barrier. These results were correlated with the inhibition of oxidative / nitrosative stress and the activity of  p38 MAPK and NF-κB [1,2]. A positive effect of CaD (2000 mg/day for 2 years) was found in patients suffering from early diabetic retinopathy [6]. Other studies failed to show a positive effect of CaD on the retina of diabetic patients [8,9].  In a recent study (CALDIRET) (30), in which patients with mild to moderate non-proliferative diabetic retinopathy were followed-up for 5 years, CaD did not reduce diabetic macular edema [7].

    1. Leal, E.C.; Martins, J.; Voabil, P.; Liberal, J.; Chiavaroli, C.; Bauer, J.; Cunha-Vaz, J.; Ambrósio, A.F. Calcium Dobesilate Inhibits the Alterations in Tight Junction Proteins and Leukocyte Adhesion to Retinal Endothelial Cells Induced by Diabetes 2010, 59(10), 2637–2645. Published online 2010 Jul 13. doi: 10.2337/db09-1421
    2. Adamis, A.P.; Berman, A.J. Immunological mechanisms in the pathogenesis of diabetic retinopathy. Semin Immunopathol. 2008, 30, 65–84. [PubMed] [Google Scholar]
    3. Kern, T.S. Contributions of inflammatory processes to the development of the early stages of diabetic retinopathy. Exp Diabetes Res. 2007, 2007, 95103. [PMC free article] [PubMed] [Google Scholar]
    4. Berthet, P.; Farine, J.C.; Barras, J.P. Calcium dobesilate: pharmacological profile related to its use in diabetic retinopathy. Int J Clin Pract. 1999, 53, 631–636. [PubMed] [Google Scholar]
    5. Tejerina, T.; Ruiz, E. Calcium dobesilate: pharmacology and future approaches. Gen Pharmacol. 1998, 31, 357–360.
    6. Ribeiro, M.L.; Seres, A.I.; Carneiro, A.M.; Stur, M.; Zourdani, A.; Caillon, P.; Cunha-Vaz, J.G. DX-Retinopathy Study Group Effect of calcium dobesilate on progression of early diabetic retinopathy: a randomised double-blind study. Graefes Arch Clin Exp Ophthalmol. 2006, 244, 1591–1600.
    7. Haritoglou, C.; Gerss, J.; Sauerland, C.; Kampik, A.; Ulbig, M.W. CALDIRET study group Effect of calcium dobesilate on occurrence of diabetic macular oedema (CALDIRET study): randomised, double-blind, placebo-controlled, multicentre trial. Lancet. 2009, 373, 1364–1371.
    8. Liu, J.; Li, S.; Sun, D. Calcium Dobesilate and Micro-vascular diseases. Life Sci. 2019, Mar 15, 221, 348-353. doi: 10.1016/j.lfs.2019.02.023. Epub 2019 Feb 12.
    9. Solà-Adell, C.; Bogdanov, P.Hernández, C.; Sampedro, J.; Valeri, M.; Garcia-Ramirez, M.; Pasquali, C.; Simó, R. Calcium Dobesilate Prevents Neurodegeneration and Vascular Leakage in Experimental Diabetes. Curr Eye Res. 2017, 42(9), 1273-1286. doi: 10.1080/02713683.2017.1302591. Epub 2017 Jun 2

    It has been found that n3-PUFA can reduce the number of retinal acellular capillaries in diabetes and inflammatory markers in the retina of diabetic animals [1,2]. Diabetic patients taking n-3 PUFA supplements for 18 months had lower risk of retinopathy [3].

    The effect of vitamin C intake on DR is not certain [4-6]. The use of vitamin C supplements may suggest that this vitamin is involved in retinopathy prevention [4].

    Millen et al. [5], conducting a prospective study of the ARIC participants,  failed to observe an effect of vitamin C and E intake from food alone or a combination of food and supplements on the risk of retinopathy. However, they found an interaction between race and vitamin E; high consumption of vitamin E from food alone, or food combined with supplements, was associated with higher prevalence of retinopathy in Caucasians, but not in African Americans [5]. Diminished likelihood of retinopathy was noted in those taking (C 3 years) vitamin C, E, and multivitamin supplements in comparison with non-users [5]. However, in a cross-sectional analysis of NHANES III participants, Millen et al. found no relationship between taking vitamin C or E supplements for a long time (C 5 vs.\1 year) and the presence of retinopathy [7].

    The intake of vitamin D and fish oil supplements was not associated with the risk of retinopathy in the ARIC [8].

    Lutein intake was not significantly correlated with retinopathy [9]. In one study, the serum level of lutein was found to be decreased in patients with nonproliferative diabetic retinopathy as compared to diabetics without retinopathy [10].

    Another study suggested a positive impact of high plasma level of lutein/zeaxanthin and lycopene on the risk of diabetic retinopathy [11]. However, increased consumption of b-carotene was found to be related to lower risk and was not associated with retinopathy [4,6].

    1. Tikhonenko, M.; Lydic, T.A.; Opreanu, M.; Li, C.S.; Bozack, S.; McSorley, K.M. et al. N-3 polyunsaturated Fatty acids prevent diabetic retinopathy by inhibition of retinal vascular damage and enhanced endothelial progenitor cell reparative function. PLoS ONE. 2013, 8(1), e55177.
    2. Shen, J.H.; Ma, Q.; Shen, S.R.; Xu, G.T.; Das, U.N. Effect of alphalinolenic acid on streptozotocin-induced diabetic retinopathy indices in vivo. Arch Med Res. 2013, 44(7), 514–520.
    3. Roig-Revert, M.J.; Lleo-Perez, A.; Zanon-Moreno, V.; Vivar-Llopis, B.; Marin-Montiel, J.; Dolz-Marco, R. et al. Enhanced oxidative stress and other potential biomarkers for retinopathy in type 2 diabetics: beneficial effects of the nutraceutic supplements. Biomed Res Int. 2015, 2015, 408180.
    4. Tanaka, S.; Yoshimura, Y.; Kawasaki, R.; Kamada, C.; Tanaka, S.; Horikawa, C. et al. Fruit intake and incident diabetic retinopathy with type 2 diabetes. 2013, 24(2), 204–201.
    5. Millen, A.E.; Klein, R.; Folsom, A.R.; Stevens, J.; Palta, M.; Mares, J.A. Relation between intake of vitamins C and E and risk of diabetic retinopathy in the Atherosclerosis Risk in Communities Study. Am J Clin Nutr. 2004, 79(5), 865–873.
    6. Mayer-Davis, E.J.; Bell, R.A.; Reboussin, B.A.; Rushing, J.; Marshall, J.A.; Hamman, R.F. Antioxidant nutrient intake and diabetic retinopathy: the San Luis Valley Diabetes Study. Ophthalmology. 1998, 105(12), 2264–2270.
    7. Millen, A.E.; Gruber, M.; Klein, R.; Klein, B.E.; Palta, M.; Mares, J.A. Relations of serum ascorbic acid and alpha-tocopherol to diabetic retinopathy in the Third National Health and Nutrition Examination Survey. Am J Epidemiol. 2003, 158(3), 225–233.
    8. Millen, A.E.; Sahli, M.W.; Nie, J.; LaMonte, M.J.; Lutsey, P.L.; Klein, B.E. et al. Adequate vitamin D status is associated with the reduced odds of prevalent diabetic retinopathy in African Americans and Caucasians. Cardiovasc Diabetol. 2016, 15(1), 128.
    9. Sahli, M.W.; Mares, J.A.; Meyers, K.J.; Klein, R.; Brady, W.E.; Klein, B.E. et al. Dietary intake of lutein and diabetic retinopathy in the atherosclerosis risk in Communities Study (ARIC). Ophthalmic Epidemiol. 2016, 23(2), 99–108.
    10. Koushan, K.; Rusovici, R. Li, W.; Ferguson, L.R.; Chalam, K.V. The role of lutein in eye-related disease. 2013, 5(5), 1823–1839.
    11. Brazionis, L.; Rowley, K.; Itsiopoulos, C.; O’Dea, K. Plasma carotenoids and diabetic retinopathy. Br J Nutr. 2009, 101(2), 270–277.

      Response to Rewiever 3

      Thank you very much for your important comments.

      Section 7.

      We have completed the information concerning the effect of coffee and tea intake on the risk of diabetic retinopathy.

      It has been shown that tea acts as a potent neuroprotector in the retina [5], preventing formation of acellular capillary vessels and pericyte ghosts in diabetic rats [1]. Green tea can protect diabetic retinal neurons and regulate the subretinal environment by decreasing  ROS generation due to increased expression of glutamate transporter, restored intercellular connections and glutamine/glutamate circulation [2]. Moreover, a low dose of green tea  is likely to improve antioxidant defense, decrease inflammatory markers and prevent thickening of the basement membrane of the retina [3]. Black tea, which lowers blood sugar and thus inhibits pathological biochemical indices, is able to delay diabetic cataract development  [4].

      As found in a Chinese case–control study, involving sex and age-matched controls suffering from diabetes but not diabetic retinopathy, regular consumption (every week for not less than 1 year) of Chinese green tea was associated with reduced probability of retinopathy in women, but not in men [6].

      1. Mustata, G.T.; Rosca, M.; Biemel, K.M.; Reihl, O.; Smith, M.A.; Viswanathan, A.; Strauch, C.; Du, Y.; Tang, J.; Kern, T.S. et al. Paradoxical effects of green tea (Camellia Sinensis) and antioxidant vitamins in diabetic rats: Improved retinopathy and renal mitochondrial defects but deterioration of collagen matrix glycoxidation and cross-linking. Diabetes. 2005, 54, 517–526. doi: 10.2337/diabetes.54.2.517. [PubMed] [CrossRef] [Google Scholar]
      2. Silva, K.C.; Rosales, M.A.B.; Hamassaki, D.E.; Saito, K.C.; Faria, A.M.; Ribeiro, P.A.O.; de Faria, J.B.L.; de Faria, J.M.L. Green tea is neuroprotective in diabetic retinopathy. Ophthalmol. Vis. Sci. 2013, 54, 1325–1336. doi: 10.1167/iovs.12-10647. [PubMed] [CrossRef] [Google Scholar]
      3. Kumar, B.; Gupta, S.K.; Nag, T.C.; Srivastava, S.; Saxena R. Green tea prevents hyperglycemia-induced retinal oxidative stress and inflammation in streptozotocin-induced diabetic rats. Ophthalmic Res. 2012, 47, 103–108. doi: 10.1159/000330051. [PubMed] [CrossRef] [Google Scholar]
      4. Vinson, J.A.; Zhang, J. Black and green teas equally inhibit diabetic cataracts in a streptozotocin-induced rat model of diabetes. Agric. Food Chem. 2005, 53, 3710–3713. doi: 10.1021/jf048052l. [PubMed] [CrossRef] [Google Scholar]
      5. Jin-Ming, Meng.; Shi-Yu, Cao.; Xin-Lin Wei, Ren-You Gan, Yuan-Feng Wang, Shu-Xian Cai, Xiao-Yu Xu, Pang-Zhen Zhang, Hua-Bin Li. Effects and Mechanisms of Tea for the Prevention and Management of Diabetes Mellitus and Diabetic Complications: An Updated Review. Antioxidants (Basel) 2019 Jun; 8(6): 170. Published online 2019 Jun 10. doi: 10.3390/antiox8060170.
      6. Ma, Q.; Chen, D.; Sun, H.P.; Yan, N.; Xu, Y.; Pan, C.W. Regular Chinese green tea consumption is protective for diabetic retinopathy: a Clinic-Based Case-Control Study. J Diabetes Res. 2015, 2015, 231570.

      Coffee is one of the most consumed beverages worldwide. The main commercial coffee blends are Arabica (Coffea arabica L.) and Robusta (Coffea canephora Pierre ex Froehner) [1]. Its main component is caffeine which belongs to antioxidants and its long-term intake in large quantities reduces oxidative stress [2]. Other components of coffee beans include carbohydrates, proteins, fats, alkaloids, diterpenes, free amino acids, melanoides and minerals, both macro- and microelements [3]. Coffee contains microelements that show antioxidant effects, such as manganese, zinc, copper and iron [1]. It can also be a source of fluorine, chromium and cobalt [1].

      A Norwegian cohort study revealed an approximately 35% reduced risk of T2D associated with high (in comparison to low) consumption of brewed coffee and other types of coffee [2]. Likewise, the results of a Finnish cohort study revealed that both the intake of boiled coffee and consumption of drip coffee were inversely correlated with T2D [3].

      A number of coffee components have been found to show anti-inflammatory effects (i.e. caffeine, CGA, cafestol, kahweol, trigonelline, caffeic and ferulic acids). Caffeine and CGA exert an effect on insulin and glucose homeostasis by modulating adenosine receptor signaling, suppressing intestinal glucose absorption (by enhanced generation of gastric inhibitory peptide-1 [GIP-1] and glucagon-like-peptide-1 [GLP-1] or glucose-6-phosphate translocase 1 inhibition), reducing glucose output in the liver (by glucose-6-phosphatase suppression), and by improving secretion of pancreatic islet insulin or peripheral insulin sensitivity and glucose uptake (by glucose transporter type 4 [GLUT4] stimulation and modulation of intracellular signaling pathway activation - Akt, AMPK, MAPK) [4].

      Regular coffee intake can lower the level of proinflammatory biomarkers, such as  L-1β, IL-6, TNF-α, C-reactive protein, monocyte chemotactic protein 1, vascular cell adhesion molecule 1, C-peptides, endothelial-leukocyte adhesion molecule 1, and interleukin 18 [IL-18]) in healthy, obese and T2D-affected individuals. However, it may increase the levels of anti-inflammatory adiponectin, interleukin 4 and interleukin 10  [5,6].

      1. Hjellvik, V.; Tverdal, A.; Strom, H. Boiled coffee intake and subsequent risk for type 2 diabetes. 2011, 22, 418–421.
      2. Olechno, E.; Pus´cion-Jakubik, A.; Socha, K.; Zujko, M.E. Coffee Infusions: Can They Be a Source of Microelements with Antioxidant Properties? Antioxidants (Basel). 2021, Oct 27, 10(11), doi: 10.3390/antiox10111709
      3. Tuomilehto, J.; Hu, G.; Bidel, Set al. , Coffee consumption and risk of type 2 diabetes mellitus among middle-aged Finnish men and women. JAMA. 2004, 291, 1213–1219.
      4. Carlström, M.; Larsson, S.C. Coffee consumption and reduced risk of developing type 2 diabetes: A systematic review with meta-analysis. Rev. 2018, 76, 395–417. [CrossRef] [PubMed]
      5. Natella, F.; Scaccini, C. Role of coffee in modulation of diabetes risk. Nutr Rev. 2012, 70, 207–217.
      6. Akash, M.S.; Rehman, K.; Chen, S. Effects of coffee on type 2 diabetes mellitus. 2014, 30, 755–763.

      Section 9.

      We have completed the information.

      Diaz-Lopez et al. studied the Mediterranean diet enhanced with extra virgin olive oil or nuts,  and compared it with a low-fat diet in more than 3600 participants in a prospective 6-year study. The Mediterranean diet enhanced with olive oil was associated with over 40% reduced risk of retinopathy [1].  The Mediterranean diet enriched with nuts was correlated with a 37% reduced, though statistically insignificant decrease in the risk of retinopathy [1].

      A Japanese cohort study of T2D individuals revealed that high fruit intake (C 173.2 g per day, i.e. an apple or two bananas), was associated with over a 50% reduction in the risk of incident retinopathy, as compared to patients who consumed less than 53.2 g of fruit daily [2].

      Milk intake has also been assessed; the consumption of whole milk and skim/low fat milk was not found to be associated with retinopathy in more than 1350 patients with type 2 diabetes [3].

      Fish may inhibit the development of retinopathy. The intake of oily fish at least twice a week (as compared to more rare consumption), was associated with a nearly 60% reduction in the risk of retinopathy [4]. Another study reported that 85–141 g of dark fish, (salmon, mackerel, swordfish, sardines, bluefish) consumed weekly versus never was related to almost 70% reduced likelihood of retinopathy [3]. However, 85–141 g of ‘‘other fish’’ (cod, perch, catfish), eaten every week was not associated with retinopathy [3].

      The intake of vitamin D and fish oil supplements, versus the non-use, was not related to the risk of retinopathy in the ARIC [3]. In an RCT, diabetics who were randomly given n-3 PUFA supplements for 18 months had a lower risk of retinopathy [5].

      Studies on the effect of fat consumption on the development of diabetic retinopathy are divided. No direct correlation was found between the intake of fat, trans fat, total saturated fatty acid (SFA) and diabetic retinopathy [6,7].

      Sasaki et al. failed to find the  impact of MUFA on retinopathy [6]. However, Alcubierre et al.  reported an inverse correlation of MUFA and oleic acid with odds of retinopathy [7].

      Some evidence indicates that the intake of PUFA may contribute to the prevention of  retinopathy [6]. However, one study found no such correlations [7].

      Considering fiber intake, two studies observed no association with retinopathy, and one Indian study found an inverse correlation [2, 7] .

      1. Diaz-Lopez A, Babio N, Martinez-Gonzalez MA, Corella D, Amor AJ, Fito M, et al. Mediterranean diet, retinopathy, nephropathy, and microvascular diabetes complications: a post hoc analysis of a randomized trial. Diabetes Care. 2015, 38(11), 2134–2141.
      2. Tanaka, S.; Yoshimura, Y.; Kawasaki, R.; Kamada, C.; Tanaka, S.; Horikawa, C. et al. Fruit intake and incident diabetic retinopathy with type 2 diabetes. 2013, 24(2), 204–211.
      3. Millen, A.E.; Sahli, M.W.; Nie, J.; LaMonte, M.J.; Lutsey, P.L.; Klein, B.E. et al. Adequate vitamin D status is associated with the reduced odds of prevalent diabetic retinopathy in African Americans and Caucasians. Cardiovasc. 2016, 15(1), 128.
      4. Sala-Vila, A.; Diaz-Lopez, A.; Valls-Pedret, C.; Cofan, M.; GarciaLayana, A.; Lamuela-Raventos, R.M. et al. Dietary Marine omega-3 fatty acids and incident sight-threatening retinopathy in middleaged and older individuals with type 2 diabetes: prospective investigation from the PREDIMED trial. JAMA Ophthalmol. 2016, 134(10), 1142–1149.
      5. Roig-Revert, M.J.; Lleo-Perez, A.; Zanon-Moreno, V.; Vivar-Llopis, B.; Marin-Montiel, J.; Dolz-Marco, R. et al. Enhanced oxidative stress and other potential biomarkers for retinopathy in type 2 diabetics: beneficial effects of the nutraceutic supplements. Biomed Res Int. 2015, 2015, 408180.
      6. Sasaki, M.; Kawasaki, R.; Rogers, S.; Man, R.E.; Itakura, K.; Xie, J. et al. The Associations of Dietary Intake of Polyunsaturated fatty acids with diabetic retinopathy in well-controlled diabetes. Invest Ophthalmol Vis Sci. 2015, 56(12), 7473–7479.
      7. Alcubierre, N.; Navarrete-Munoz, E.M.; Rubinat, E.; Falguera, M.; Valls, J.; Traveset, A. et al. Association of low oleic acid intake with diabetic retinopathy in type 2 diabetic patients: a casecontrol study. Nutr Metab (Lond). 2016, 13, 40.

      Section 8.

      We have corrected manucrypt according to the recommendations.

      It has been shown that DM patients leading a sedentary life style have a higher risk of diabetic retinopathy as compared to those living actively [1]. Less physically active diabetic patients showed increased blood flow in the retina on exertion [2,3].

      Meta-analysis by Umpierre et al. showed that more structured training, following the ADA’s guideline (> 150 min per week), and receiving PA advice alone were correlated with a greater reduction in HbA1c in T2DM patients [4]. 

      Meta-analysis by Boniol et al. also suggested a possible mechanism of PA’s impact on DR due to glycemic control improvement [5]. Alteration in the 25-hydroxyvitamin D (25OH-D) level can be another likely mechanism. Substantial evidence has shown that higher PA level contributes to the improved 25OH-D status in people at all ages [6-10]. Keech et al. observed lower blood 25OH-D concentration in relation to a higher risk of macrovascular and microvascular events, including DR [11].

      It has been proved that physical exercise is able to modulate oxidative stress [12]. A few experiments have revealed a decrease in oxidative stress in the retinas of DR mice during physical exercise [13-16], and a positive alterations in microglia in rats with streptozotocin-induced DM following treadmill exercise [17].

      It has to be remembered, however, that patients suffering from proliferative diabetic retinopathy should avoid high intensity aerobic and resistance exercise to reduce the risk of vitreous hemorrhage or retinal detachment [18,19]. All types of physical exercise that may predispose to elevated systolic blood pressure (Valsalva maneuvers) increase the risk of vitreous hemorrhage [20,21].

      1. Ren, C.; Liu, W.; Li, J.; Cao, Y.; Xu, J.; Lu, P. Physical activity and risk of diabetic retinopathy: A systematic review and meta-analysis. Acta Diabetol. 2019, 56(8), 823–37. doi: 10.1007/s00592-019-01319-4. - DOI – PubMed
      2. Hayashi, N.; Ikemura, T.; Someya, N. Effects of dynamic exercise and its intensity on ocular blood flow in humans. Eur J Appl Physiol. 2011, 111(10), 2601–2606.
      3. Zhang, Y.; San Emeterio Nateras, O.; Peng, Q. et al. Blood flow MRI of the human retina/choroid during rest and isometric exercise. Invest Ophthalmol Vis Sci. 2012, 53(7), 4299–4305.
      4. Umpierre, D.; Ribeiro, P.A.; Kramer, C.K. et al. Physical activity advice only or structured exercise training and association with HbA1c levels in type 2 diabetes: a systematic review and meta-analysis. JAMA. 2011, 305(17), 1790–1799.
      5. Boniol, M.; Dragomir, M. Physical activity and change in fasting glucose and HbA1c: a quantitative meta-analysis of randomized trials. Acta Diabetol. 2017, 54(11), 983–991.
      6. Al-Othman, A.; Al-Musharaf, S.; Al-Daghri, N.M. et al. Effect of physical activity and sun exposure on vitamin D status of Saudi children and adolescents. BMC Pediatr. 2012, 12, 92.
      7. Scott, D.; Blizzard, L.; Fell, J. et al. A prospective study of the associations between 25-hydroxy-vitamin D, sarcopenia progression and physical activity in older adults. Clin Endocrinol (Oxf). 2010, 3(5), 581–587.
      8. Klenk, J.; Rapp, K.; Denkinger, M et al. Objectively measured physical activity and vitamin D status in older people from Germany. J Epidemiol Community Health. 2015, 69(4), 388–392.
      9. Makanae, Y.; Ogasawara, R.; Sato, K. et al. Acute bout of resistance exercise increases vitamin D receptor protein expression in rat skeletal muscle. Exp Physiol. 2015, 100(10), 1168–1176.
      10. Black, L.J.; Burrows, S.A., Jacoby, P. et al. Vitamin D status and predictors of serum 25-hydroxyvitamin D concentrations in Western Australian adolescents. Br J Nutr. 2014, 112(7), 1154–1162.
      11. Keech, A.C., Mitchell, P.; Summanen, P.A. et al. Effect of fenofibrate on the need for laser treatment for diabetic retinopathy (FIELD study): a randomised controlled trial. Lancet. 2007, 370(9600), 1687–1697.
      12. Sallam, N.; Laher, I. Exercise modulates oxidative stress and inflammation in aging and cardiovascular diseases. Oxid Med Cell Longev. 2016, 2016, 7239639.
      13. Kim, C.S.; Park, S.; Chun, Y. et al. Treadmill exercise attenuates retinal oxidative stress in naturally-aged mice: an immunohistochemical study. Int J Mol Sci. 2015, 16(9), 21008–21020.
      14. Kruk, J.; Kubasik-Kladna, K.; Aboul-Enein, H.Y. The role oxidative stress in the pathogenesis of eye diseases: current status and a dual role of physical activity. Mini Rev Med Chem. 2015, 16(3), 241–257.
      15. Allen, R.S.; Hanif, A.M.; Gogniat, M.A. et al. TrkB signalling pathway mediates the protective effects of exercise in the diabetic rat retina. Eur J Neurosci. 2018, 47(10), 1254–1265.
      16. Cui, J.Z.; Wong, M.; Wang, A. et al. Exercise inhibits progression of diabetic retinopathy by reducing inflammatory, oxidative stress, and ER stress gene expression in the retina of db/db mice. Invest Ophthalmol Vis Sci. 2016, 57(12), 5434.
      17. Lu, Y.; Dong, Y.; Tucker, D. et al. Treadmill exercise exerts neuroprotection and regulates microglial polarization and oxidative stress in a streptozotocin-induced rat model of sporadic alzheimer’s disease. J Alzheimers Dis. 2017, 56(4), 1469–1484.
      18. Schneider, S.H.; Khachadurian, A.K.; Amorosa, L.F. et al. Tenyear experience with an exercise-based outpatient life-style modification program in the treatment of diabetes mellitus. Diabetes Care. 1992, 15(11), 1800–1810.
      19. Colberg, S.R. Exercise and diabetes: a clinician’s guide to prescribing physical activity, 1st edn. American Diabetes Association, 2013. Alexandria.
      20. Graham, C.; Lasko-McCarthey, P. Exercise options for persons with diabetic complications. Diabetes Educ. 1990, 16(3), 212–220.
      21. Hamdy, O.; Goodyear, L.J.; Horton, E.S. Diet and exercise in type 2 diabetes mellitus. Endocrinol Metab Clin North Am. 2001, 30(4), 883–907.

      Section 8.

      We have completed the information.

      Research using animal models of diabetes has shown that resistance exercise can cause increased muscle mass [1]. The skeletal muscle is a substantial glucose reservoir in the body and physical exercise is a potent stimulant of glucose uptake partly via the skeletal muscle glucose transporter protein action [2]. That is why resistance training exerting a direct effect on skeletal muscle is likely to play a role in the management of D2T patients [3].

      Resistance training is a good alternative to aerobic exercise. This type of training has not been found to cause a greater number of undesirable events as compared to other types of exercise [3]. Pollock et al. reported that there were more complications during jogging and walking than in strength training in the elderly [4].

      Compliance with recommendations related to aerobic exercise can be difficult. Aerobic training should be conducted almost every day or even all the week to be effective. However, progressive resistance exercise can be effective if performed only three times a week [3].

      Moreover, some coexisting diseases in diabetic patients (cardiovascular and peripheral vascular disorders, neuropathy and motor impairment – foot ulceration, claudication and a risk of falling down) may hinder aerobic performance. In all these situations, resistance training is not only a real alternative but can also be more viable than aerobic exercise. In patients who can safely take aerobic exercise, a combined training program is more effective than the progressive resistance training program alone [3]. Sigal et al. reported that the levels of glycosylated hemoglobin decreased markedly in the combined training group as compared to the only aerobic or resistance training groups [5].

      Eight weeks with two or three 45 min. sessions of progressive training are sufficient to improve glycemic control [3].

      A reduction in glycosylated hemoglobin by 1% is associated with a 37% decrease in the risk of microvascular complications and a 21% decrease in the risk of diabetes-related death [6].

      Snowling and Hopkins observed a 0.5% reduction in glycosylated hemoglobin after progressive resistance training [7].

      1. Farrell, P.; Fedele, M.; Hernandez, J.; Fluckey, J.; Miller, J.; Lang, C. et al. Hypertrophy of skeletal muscle in diabetic rats in response to chronic resistance exercise. Journal of Applied Physiology. 1999, 87, 1075–1082.
      2. Schuller, G.; Linke, A. Diabetes, exercise. In Goldstein, B.; Muller-Wieland, D. (Eds) Type 2 Diabetes: Principles, Practice. New York: Informa Healthcare, Ch 6. 2008
      3. Irvine, C.; Taylor, N.C. Progressive resistance exercise improves glycaemic control in people with type 2 diabetes mellitus: a systematic review. Australian Journal of Physiotherapy. 2009, 55, 237-246.
      4. Pollock, M.; Carroll, J.; Graves, J.; Leggett, S.; Braith, R.; Limacher, M. et al. Injuries, adherence to walk/jog, resistance training programs in the elderly. Medicine, Science in Sports, Exercise. 1991, 23, 1194–1200.
      5. Sigal, R.; Kenny, G.; Boule, N.; Wells, G.; Prud’homme, D.; Fortier, M. et al. Effects of aerobic training, resistance training, or both on glycemic control in type 2 diabetes: A randomized trial. Annals of Internal Medicine. 2007, 147, 357–369.
      6. Stratton, I.; Adler, A.; Neil, H.; Matthews, D.; Manley, S.; Cull, C. et al. Association of glycemia with macrovascular, microvascular complications of type 2 diabetes (UKPDS35): Prospective observational study. BMJ. 2000, 321, 405–412.
      7. Snowling, N.; Hopkins, W. Effects of different modes of exercise training on glucose control, risk factors for complications in type 2 diabetic patients: a meta-analysis. Diabetes Care. 2006, 29, 2518–2527.

      Section 11.

      Dodaliśmy zalecane informacje.

      Chlorogenic acid (CGA) is a polyphenol found in a variety of products, such as coffee, grains, potatoes and apples. It is the ester of caffeic and quinic acids, showing antibacterial, antiinflammatory, antioxidant and anineoplastic effects [1-3]. CGA can delay glucose absorption in the small intestine and decrease glucose production in the liver [4,5]. It is believed that CGA stimulates secretion of glucagon-like peptide 1 known to exert a beneficial effect on the response to glucose in pancreatic beta cells [3]. In the liver, CGA inhibits glucose-6-phospatase, thus decreasing hepatic glucose production [5,6]. CGA reduces hyperpermeability of retinal vessels in diabetic rats. The diabetic rats with higher level of VEGF and down-regulation of occludin, claudin-5 and ZO-1 showed breakdown of the blood-retina barrier and intensified vascular leakage. CGA managed to maintain occludin expression and reduced the levels of VEGF, which decreased the BRB breakdown and inhibited vascular leakage. Thus, CGA may prevent BRB breakdown in retinopathy [7].

      1. Dos Santos, M.D.; Almeida, M.C.; Lopes, N.P.; de Souza, G.E. Evaluation of the anti-inflammatory, analgesic and antipyretic activities of the natural polyphenol chlorogenic acid. Biol Pharm Bull. 2006, 29, 2236-2240.
      2. Puupponen-Pimiä, R.; Nohynek, L.; Meier, C.; Kähkönen, M.; Heinonen, M.; Hopia, A, Oksman-Caldentey, K.M. Antimicrobial properties of phenolic compounds from berries. J Appl Microbiol. 2001, 90, 494-507.
      3. Ma, C.M.; Kully, M.; Khan, J.K.; Hattori, M.; Daneshtalab, M. Synthesis of chlorogenic acid derivatives with promising antifungal activity. Bioorg Med Chem. 2007, 15, 6830-6833.
      4. McCarty, M.F. A chlorogenic acid-induced increase in GLP-1 production may mediate the impact of heavy coffee consumption on diabetes risk. Med Hypotheses. 2005, 64, 848-853.
      5. Arion, W.J.; Canfield, W.K.; Ramos, F.C. Schindler, P.W.; Burger, H.J.; Hemmerle, H.; Schubert, G.; Below, P.; Herling, AW. Chlorogenic acid and hydroxynitrobenzaldehyde: new inhibitors of hepatic glucose 6-phosphatase. Arch Biochem Biophys. 1997, 339, 315-322.
      6. Herling, A.W.; Burger, H.; Schubert, G.; Hemmerle, H.; Schaefer, H.; Kramer, W. Alterations of carbohydrate and lipid intermediary metabolism during inhibition of glucose-6-phosphatase in rats. Eur J Pharmacol. 1999, 386, 75-82.
      7. Shin, J.Y.; Sohn, J.; Hyung, K. ParkChlorogenic Acid Decreases Retinal Vascular Hyperpermeability in Diabetic Rat Model. Korean Med Sci .2013, 28, 608-613.

      Calcium dobesilate (CaD) is considered to be an angioprotective drug. For many years CaD was used in diabetic retinopathy [4,5]. Researchers investigating its efficacy are divided. A few clinical studies showed delayed progression of diabetic retinopathy after long-term oral treatment with CaD. It was found to inhibit changes in tight junction proteins, ICAM-1 and leukocyte adhesion to retinal vessels which are known to lie at the base of growing permeability of the blood-retina barrier. These results were correlated with the inhibition of oxidative / nitrosative stress and the activity of  p38 MAPK and NF-κB [1,2]. A positive effect of CaD (2000 mg/day for 2 years) was found in patients suffering from early diabetic retinopathy [6]. Other studies failed to show a positive effect of CaD on the retina of diabetic patients [8,9].  In a recent study (CALDIRET) (30), in which patients with mild to moderate non-proliferative diabetic retinopathy were followed-up for 5 years, CaD did not reduce diabetic macular edema [7].

      1. Leal, E.C.; Martins, J.; Voabil, P.; Liberal, J.; Chiavaroli, C.; Bauer, J.; Cunha-Vaz, J.; Ambrósio, A.F. Calcium Dobesilate Inhibits the Alterations in Tight Junction Proteins and Leukocyte Adhesion to Retinal Endothelial Cells Induced by Diabetes 2010, 59(10), 2637–2645. Published online 2010 Jul 13. doi: 10.2337/db09-1421
      2. Adamis, A.P.; Berman, A.J. Immunological mechanisms in the pathogenesis of diabetic retinopathy. Semin Immunopathol. 2008, 30, 65–84. [PubMed] [Google Scholar]
      3. Kern, T.S. Contributions of inflammatory processes to the development of the early stages of diabetic retinopathy. Exp Diabetes Res. 2007, 2007, 95103. [PMC free article] [PubMed] [Google Scholar]
      4. Berthet, P.; Farine, J.C.; Barras, J.P. Calcium dobesilate: pharmacological profile related to its use in diabetic retinopathy. Int J Clin Pract. 1999, 53, 631–636. [PubMed] [Google Scholar]
      5. Tejerina, T.; Ruiz, E. Calcium dobesilate: pharmacology and future approaches. Gen Pharmacol. 1998, 31, 357–360.
      6. Ribeiro, M.L.; Seres, A.I.; Carneiro, A.M.; Stur, M.; Zourdani, A.; Caillon, P.; Cunha-Vaz, J.G. DX-Retinopathy Study Group Effect of calcium dobesilate on progression of early diabetic retinopathy: a randomised double-blind study. Graefes Arch Clin Exp Ophthalmol. 2006, 244, 1591–1600.
      7. Haritoglou, C.; Gerss, J.; Sauerland, C.; Kampik, A.; Ulbig, M.W. CALDIRET study group Effect of calcium dobesilate on occurrence of diabetic macular oedema (CALDIRET study): randomised, double-blind, placebo-controlled, multicentre trial. Lancet. 2009, 373, 1364–1371.
      8. Liu, J.; Li, S.; Sun, D. Calcium Dobesilate and Micro-vascular diseases. Life Sci. 2019, Mar 15, 221, 348-353. doi: 10.1016/j.lfs.2019.02.023. Epub 2019 Feb 12.
      9. Solà-Adell, C.; Bogdanov, P.Hernández, C.; Sampedro, J.; Valeri, M.; Garcia-Ramirez, M.; Pasquali, C.; Simó, R. Calcium Dobesilate Prevents Neurodegeneration and Vascular Leakage in Experimental Diabetes. Curr Eye Res. 2017, 42(9), 1273-1286. doi: 10.1080/02713683.2017.1302591. Epub 2017 Jun 2

      It has been found that n3-PUFA can reduce the number of retinal acellular capillaries in diabetes and inflammatory markers in the retina of diabetic animals [1,2]. Diabetic patients taking n-3 PUFA supplements for 18 months had lower risk of retinopathy [3].

      The effect of vitamin C intake on DR is not certain [4-6]. The use of vitamin C supplements may suggest that this vitamin is involved in retinopathy prevention [4].

      Millen et al. [5], conducting a prospective study of the ARIC participants,  failed to observe an effect of vitamin C and E intake from food alone or a combination of food and supplements on the risk of retinopathy. However, they found an interaction between race and vitamin E; high consumption of vitamin E from food alone, or food combined with supplements, was associated with higher prevalence of retinopathy in Caucasians, but not in African Americans [5]. Diminished likelihood of retinopathy was noted in those taking (C 3 years) vitamin C, E, and multivitamin supplements in comparison with non-users [5]. However, in a cross-sectional analysis of NHANES III participants, Millen et al. found no relationship between taking vitamin C or E supplements for a long time (C 5 vs.\1 year) and the presence of retinopathy [7].

      The intake of vitamin D and fish oil supplements was not associated with the risk of retinopathy in the ARIC [8].

      Lutein intake was not significantly correlated with retinopathy [9]. In one study, the serum level of lutein was found to be decreased in patients with nonproliferative diabetic retinopathy as compared to diabetics without retinopathy [10].

      Another study suggested a positive impact of high plasma level of lutein/zeaxanthin and lycopene on the risk of diabetic retinopathy [11]. However, increased consumption of b-carotene was found to be related to lower risk and was not associated with retinopathy [4,6].

      1. Tikhonenko, M.; Lydic, T.A.; Opreanu, M.; Li, C.S.; Bozack, S.; McSorley, K.M. et al. N-3 polyunsaturated Fatty acids prevent diabetic retinopathy by inhibition of retinal vascular damage and enhanced endothelial progenitor cell reparative function. PLoS ONE. 2013, 8(1), e55177.
      2. Shen, J.H.; Ma, Q.; Shen, S.R.; Xu, G.T.; Das, U.N. Effect of alphalinolenic acid on streptozotocin-induced diabetic retinopathy indices in vivo. Arch Med Res. 2013, 44(7), 514–520.
      3. Roig-Revert, M.J.; Lleo-Perez, A.; Zanon-Moreno, V.; Vivar-Llopis, B.; Marin-Montiel, J.; Dolz-Marco, R. et al. Enhanced oxidative stress and other potential biomarkers for retinopathy in type 2 diabetics: beneficial effects of the nutraceutic supplements. Biomed Res Int. 2015, 2015, 408180.
      4. Tanaka, S.; Yoshimura, Y.; Kawasaki, R.; Kamada, C.; Tanaka, S.; Horikawa, C. et al. Fruit intake and incident diabetic retinopathy with type 2 diabetes. 2013, 24(2), 204–201.
      5. Millen, A.E.; Klein, R.; Folsom, A.R.; Stevens, J.; Palta, M.; Mares, J.A. Relation between intake of vitamins C and E and risk of diabetic retinopathy in the Atherosclerosis Risk in Communities Study. Am J Clin Nutr. 2004, 79(5), 865–873.
      6. Mayer-Davis, E.J.; Bell, R.A.; Reboussin, B.A.; Rushing, J.; Marshall, J.A.; Hamman, R.F. Antioxidant nutrient intake and diabetic retinopathy: the San Luis Valley Diabetes Study. Ophthalmology. 1998, 105(12), 2264–2270.
      7. Millen, A.E.; Gruber, M.; Klein, R.; Klein, B.E.; Palta, M.; Mares, J.A. Relations of serum ascorbic acid and alpha-tocopherol to diabetic retinopathy in the Third National Health and Nutrition Examination Survey. Am J Epidemiol. 2003, 158(3), 225–233.
      8. Millen, A.E.; Sahli, M.W.; Nie, J.; LaMonte, M.J.; Lutsey, P.L.; Klein, B.E. et al. Adequate vitamin D status is associated with the reduced odds of prevalent diabetic retinopathy in African Americans and Caucasians. Cardiovasc Diabetol. 2016, 15(1), 128.
      9. Sahli, M.W.; Mares, J.A.; Meyers, K.J.; Klein, R.; Brady, W.E.; Klein, B.E. et al. Dietary intake of lutein and diabetic retinopathy in the atherosclerosis risk in Communities Study (ARIC). Ophthalmic Epidemiol. 2016, 23(2), 99–108.
      10. Koushan, K.; Rusovici, R. Li, W.; Ferguson, L.R.; Chalam, K.V. The role of lutein in eye-related disease. 2013, 5(5), 1823–1839.
      11. Brazionis, L.; Rowley, K.; Itsiopoulos, C.; O’Dea, K. Plasma carotenoids and diabetic retinopathy. Br J Nutr. 2009, 101(2), 270–277.

        Response to Rewiever 3

        Thank you very much for your important comments.

        Section 7.

        We have completed the information concerning the effect of coffee and tea intake on the risk of diabetic retinopathy.

        It has been shown that tea acts as a potent neuroprotector in the retina [5], preventing formation of acellular capillary vessels and pericyte ghosts in diabetic rats [1]. Green tea can protect diabetic retinal neurons and regulate the subretinal environment by decreasing  ROS generation due to increased expression of glutamate transporter, restored intercellular connections and glutamine/glutamate circulation [2]. Moreover, a low dose of green tea  is likely to improve antioxidant defense, decrease inflammatory markers and prevent thickening of the basement membrane of the retina [3]. Black tea, which lowers blood sugar and thus inhibits pathological biochemical indices, is able to delay diabetic cataract development  [4].

        As found in a Chinese case–control study, involving sex and age-matched controls suffering from diabetes but not diabetic retinopathy, regular consumption (every week for not less than 1 year) of Chinese green tea was associated with reduced probability of retinopathy in women, but not in men [6].

        1. Mustata, G.T.; Rosca, M.; Biemel, K.M.; Reihl, O.; Smith, M.A.; Viswanathan, A.; Strauch, C.; Du, Y.; Tang, J.; Kern, T.S. et al. Paradoxical effects of green tea (Camellia Sinensis) and antioxidant vitamins in diabetic rats: Improved retinopathy and renal mitochondrial defects but deterioration of collagen matrix glycoxidation and cross-linking. Diabetes. 2005, 54, 517–526. doi: 10.2337/diabetes.54.2.517. [PubMed] [CrossRef] [Google Scholar]
        2. Silva, K.C.; Rosales, M.A.B.; Hamassaki, D.E.; Saito, K.C.; Faria, A.M.; Ribeiro, P.A.O.; de Faria, J.B.L.; de Faria, J.M.L. Green tea is neuroprotective in diabetic retinopathy. Ophthalmol. Vis. Sci. 2013, 54, 1325–1336. doi: 10.1167/iovs.12-10647. [PubMed] [CrossRef] [Google Scholar]
        3. Kumar, B.; Gupta, S.K.; Nag, T.C.; Srivastava, S.; Saxena R. Green tea prevents hyperglycemia-induced retinal oxidative stress and inflammation in streptozotocin-induced diabetic rats. Ophthalmic Res. 2012, 47, 103–108. doi: 10.1159/000330051. [PubMed] [CrossRef] [Google Scholar]
        4. Vinson, J.A.; Zhang, J. Black and green teas equally inhibit diabetic cataracts in a streptozotocin-induced rat model of diabetes. Agric. Food Chem. 2005, 53, 3710–3713. doi: 10.1021/jf048052l. [PubMed] [CrossRef] [Google Scholar]
        5. Jin-Ming, Meng.; Shi-Yu, Cao.; Xin-Lin Wei, Ren-You Gan, Yuan-Feng Wang, Shu-Xian Cai, Xiao-Yu Xu, Pang-Zhen Zhang, Hua-Bin Li. Effects and Mechanisms of Tea for the Prevention and Management of Diabetes Mellitus and Diabetic Complications: An Updated Review. Antioxidants (Basel) 2019 Jun; 8(6): 170. Published online 2019 Jun 10. doi: 10.3390/antiox8060170.
        6. Ma, Q.; Chen, D.; Sun, H.P.; Yan, N.; Xu, Y.; Pan, C.W. Regular Chinese green tea consumption is protective for diabetic retinopathy: a Clinic-Based Case-Control Study. J Diabetes Res. 2015, 2015, 231570.

        Coffee is one of the most consumed beverages worldwide. The main commercial coffee blends are Arabica (Coffea arabica L.) and Robusta (Coffea canephora Pierre ex Froehner) [1]. Its main component is caffeine which belongs to antioxidants and its long-term intake in large quantities reduces oxidative stress [2]. Other components of coffee beans include carbohydrates, proteins, fats, alkaloids, diterpenes, free amino acids, melanoides and minerals, both macro- and microelements [3]. Coffee contains microelements that show antioxidant effects, such as manganese, zinc, copper and iron [1]. It can also be a source of fluorine, chromium and cobalt [1].

        A Norwegian cohort study revealed an approximately 35% reduced risk of T2D associated with high (in comparison to low) consumption of brewed coffee and other types of coffee [2]. Likewise, the results of a Finnish cohort study revealed that both the intake of boiled coffee and consumption of drip coffee were inversely correlated with T2D [3].

        A number of coffee components have been found to show anti-inflammatory effects (i.e. caffeine, CGA, cafestol, kahweol, trigonelline, caffeic and ferulic acids). Caffeine and CGA exert an effect on insulin and glucose homeostasis by modulating adenosine receptor signaling, suppressing intestinal glucose absorption (by enhanced generation of gastric inhibitory peptide-1 [GIP-1] and glucagon-like-peptide-1 [GLP-1] or glucose-6-phosphate translocase 1 inhibition), reducing glucose output in the liver (by glucose-6-phosphatase suppression), and by improving secretion of pancreatic islet insulin or peripheral insulin sensitivity and glucose uptake (by glucose transporter type 4 [GLUT4] stimulation and modulation of intracellular signaling pathway activation - Akt, AMPK, MAPK) [4].

        Regular coffee intake can lower the level of proinflammatory biomarkers, such as  L-1β, IL-6, TNF-α, C-reactive protein, monocyte chemotactic protein 1, vascular cell adhesion molecule 1, C-peptides, endothelial-leukocyte adhesion molecule 1, and interleukin 18 [IL-18]) in healthy, obese and T2D-affected individuals. However, it may increase the levels of anti-inflammatory adiponectin, interleukin 4 and interleukin 10  [5,6].

        1. Hjellvik, V.; Tverdal, A.; Strom, H. Boiled coffee intake and subsequent risk for type 2 diabetes. 2011, 22, 418–421.
        2. Olechno, E.; Pus´cion-Jakubik, A.; Socha, K.; Zujko, M.E. Coffee Infusions: Can They Be a Source of Microelements with Antioxidant Properties? Antioxidants (Basel). 2021, Oct 27, 10(11), doi: 10.3390/antiox10111709
        3. Tuomilehto, J.; Hu, G.; Bidel, Set al. , Coffee consumption and risk of type 2 diabetes mellitus among middle-aged Finnish men and women. JAMA. 2004, 291, 1213–1219.
        4. Carlström, M.; Larsson, S.C. Coffee consumption and reduced risk of developing type 2 diabetes: A systematic review with meta-analysis. Rev. 2018, 76, 395–417. [CrossRef] [PubMed]
        5. Natella, F.; Scaccini, C. Role of coffee in modulation of diabetes risk. Nutr Rev. 2012, 70, 207–217.
        6. Akash, M.S.; Rehman, K.; Chen, S. Effects of coffee on type 2 diabetes mellitus. 2014, 30, 755–763.

        Section 9.

        We have completed the information.

        Diaz-Lopez et al. studied the Mediterranean diet enhanced with extra virgin olive oil or nuts,  and compared it with a low-fat diet in more than 3600 participants in a prospective 6-year study. The Mediterranean diet enhanced with olive oil was associated with over 40% reduced risk of retinopathy [1].  The Mediterranean diet enriched with nuts was correlated with a 37% reduced, though statistically insignificant decrease in the risk of retinopathy [1].

        A Japanese cohort study of T2D individuals revealed that high fruit intake (C 173.2 g per day, i.e. an apple or two bananas), was associated with over a 50% reduction in the risk of incident retinopathy, as compared to patients who consumed less than 53.2 g of fruit daily [2].

        Milk intake has also been assessed; the consumption of whole milk and skim/low fat milk was not found to be associated with retinopathy in more than 1350 patients with type 2 diabetes [3].

        Fish may inhibit the development of retinopathy. The intake of oily fish at least twice a week (as compared to more rare consumption), was associated with a nearly 60% reduction in the risk of retinopathy [4]. Another study reported that 85–141 g of dark fish, (salmon, mackerel, swordfish, sardines, bluefish) consumed weekly versus never was related to almost 70% reduced likelihood of retinopathy [3]. However, 85–141 g of ‘‘other fish’’ (cod, perch, catfish), eaten every week was not associated with retinopathy [3].

        The intake of vitamin D and fish oil supplements, versus the non-use, was not related to the risk of retinopathy in the ARIC [3]. In an RCT, diabetics who were randomly given n-3 PUFA supplements for 18 months had a lower risk of retinopathy [5].

        Studies on the effect of fat consumption on the development of diabetic retinopathy are divided. No direct correlation was found between the intake of fat, trans fat, total saturated fatty acid (SFA) and diabetic retinopathy [6,7].

        Sasaki et al. failed to find the  impact of MUFA on retinopathy [6]. However, Alcubierre et al.  reported an inverse correlation of MUFA and oleic acid with odds of retinopathy [7].

        Some evidence indicates that the intake of PUFA may contribute to the prevention of  retinopathy [6]. However, one study found no such correlations [7].

        Considering fiber intake, two studies observed no association with retinopathy, and one Indian study found an inverse correlation [2, 7] .

        1. Diaz-Lopez A, Babio N, Martinez-Gonzalez MA, Corella D, Amor AJ, Fito M, et al. Mediterranean diet, retinopathy, nephropathy, and microvascular diabetes complications: a post hoc analysis of a randomized trial. Diabetes Care. 2015, 38(11), 2134–2141.
        2. Tanaka, S.; Yoshimura, Y.; Kawasaki, R.; Kamada, C.; Tanaka, S.; Horikawa, C. et al. Fruit intake and incident diabetic retinopathy with type 2 diabetes. 2013, 24(2), 204–211.
        3. Millen, A.E.; Sahli, M.W.; Nie, J.; LaMonte, M.J.; Lutsey, P.L.; Klein, B.E. et al. Adequate vitamin D status is associated with the reduced odds of prevalent diabetic retinopathy in African Americans and Caucasians. Cardiovasc. 2016, 15(1), 128.
        4. Sala-Vila, A.; Diaz-Lopez, A.; Valls-Pedret, C.; Cofan, M.; GarciaLayana, A.; Lamuela-Raventos, R.M. et al. Dietary Marine omega-3 fatty acids and incident sight-threatening retinopathy in middleaged and older individuals with type 2 diabetes: prospective investigation from the PREDIMED trial. JAMA Ophthalmol. 2016, 134(10), 1142–1149.
        5. Roig-Revert, M.J.; Lleo-Perez, A.; Zanon-Moreno, V.; Vivar-Llopis, B.; Marin-Montiel, J.; Dolz-Marco, R. et al. Enhanced oxidative stress and other potential biomarkers for retinopathy in type 2 diabetics: beneficial effects of the nutraceutic supplements. Biomed Res Int. 2015, 2015, 408180.
        6. Sasaki, M.; Kawasaki, R.; Rogers, S.; Man, R.E.; Itakura, K.; Xie, J. et al. The Associations of Dietary Intake of Polyunsaturated fatty acids with diabetic retinopathy in well-controlled diabetes. Invest Ophthalmol Vis Sci. 2015, 56(12), 7473–7479.
        7. Alcubierre, N.; Navarrete-Munoz, E.M.; Rubinat, E.; Falguera, M.; Valls, J.; Traveset, A. et al. Association of low oleic acid intake with diabetic retinopathy in type 2 diabetic patients: a casecontrol study. Nutr Metab (Lond). 2016, 13, 40.

        Section 8.

        We have corrected manucrypt according to the recommendations.

        It has been shown that DM patients leading a sedentary life style have a higher risk of diabetic retinopathy as compared to those living actively [1]. Less physically active diabetic patients showed increased blood flow in the retina on exertion [2,3].

        Meta-analysis by Umpierre et al. showed that more structured training, following the ADA’s guideline (> 150 min per week), and receiving PA advice alone were correlated with a greater reduction in HbA1c in T2DM patients [4]. 

        Meta-analysis by Boniol et al. also suggested a possible mechanism of PA’s impact on DR due to glycemic control improvement [5]. Alteration in the 25-hydroxyvitamin D (25OH-D) level can be another likely mechanism. Substantial evidence has shown that higher PA level contributes to the improved 25OH-D status in people at all ages [6-10]. Keech et al. observed lower blood 25OH-D concentration in relation to a higher risk of macrovascular and microvascular events, including DR [11].

        It has been proved that physical exercise is able to modulate oxidative stress [12]. A few experiments have revealed a decrease in oxidative stress in the retinas of DR mice during physical exercise [13-16], and a positive alterations in microglia in rats with streptozotocin-induced DM following treadmill exercise [17].

        It has to be remembered, however, that patients suffering from proliferative diabetic retinopathy should avoid high intensity aerobic and resistance exercise to reduce the risk of vitreous hemorrhage or retinal detachment [18,19]. All types of physical exercise that may predispose to elevated systolic blood pressure (Valsalva maneuvers) increase the risk of vitreous hemorrhage [20,21].

        1. Ren, C.; Liu, W.; Li, J.; Cao, Y.; Xu, J.; Lu, P. Physical activity and risk of diabetic retinopathy: A systematic review and meta-analysis. Acta Diabetol. 2019, 56(8), 823–37. doi: 10.1007/s00592-019-01319-4. - DOI – PubMed
        2. Hayashi, N.; Ikemura, T.; Someya, N. Effects of dynamic exercise and its intensity on ocular blood flow in humans. Eur J Appl Physiol. 2011, 111(10), 2601–2606.
        3. Zhang, Y.; San Emeterio Nateras, O.; Peng, Q. et al. Blood flow MRI of the human retina/choroid during rest and isometric exercise. Invest Ophthalmol Vis Sci. 2012, 53(7), 4299–4305.
        4. Umpierre, D.; Ribeiro, P.A.; Kramer, C.K. et al. Physical activity advice only or structured exercise training and association with HbA1c levels in type 2 diabetes: a systematic review and meta-analysis. JAMA. 2011, 305(17), 1790–1799.
        5. Boniol, M.; Dragomir, M. Physical activity and change in fasting glucose and HbA1c: a quantitative meta-analysis of randomized trials. Acta Diabetol. 2017, 54(11), 983–991.
        6. Al-Othman, A.; Al-Musharaf, S.; Al-Daghri, N.M. et al. Effect of physical activity and sun exposure on vitamin D status of Saudi children and adolescents. BMC Pediatr. 2012, 12, 92.
        7. Scott, D.; Blizzard, L.; Fell, J. et al. A prospective study of the associations between 25-hydroxy-vitamin D, sarcopenia progression and physical activity in older adults. Clin Endocrinol (Oxf). 2010, 3(5), 581–587.
        8. Klenk, J.; Rapp, K.; Denkinger, M et al. Objectively measured physical activity and vitamin D status in older people from Germany. J Epidemiol Community Health. 2015, 69(4), 388–392.
        9. Makanae, Y.; Ogasawara, R.; Sato, K. et al. Acute bout of resistance exercise increases vitamin D receptor protein expression in rat skeletal muscle. Exp Physiol. 2015, 100(10), 1168–1176.
        10. Black, L.J.; Burrows, S.A., Jacoby, P. et al. Vitamin D status and predictors of serum 25-hydroxyvitamin D concentrations in Western Australian adolescents. Br J Nutr. 2014, 112(7), 1154–1162.
        11. Keech, A.C., Mitchell, P.; Summanen, P.A. et al. Effect of fenofibrate on the need for laser treatment for diabetic retinopathy (FIELD study): a randomised controlled trial. Lancet. 2007, 370(9600), 1687–1697.
        12. Sallam, N.; Laher, I. Exercise modulates oxidative stress and inflammation in aging and cardiovascular diseases. Oxid Med Cell Longev. 2016, 2016, 7239639.
        13. Kim, C.S.; Park, S.; Chun, Y. et al. Treadmill exercise attenuates retinal oxidative stress in naturally-aged mice: an immunohistochemical study. Int J Mol Sci. 2015, 16(9), 21008–21020.
        14. Kruk, J.; Kubasik-Kladna, K.; Aboul-Enein, H.Y. The role oxidative stress in the pathogenesis of eye diseases: current status and a dual role of physical activity. Mini Rev Med Chem. 2015, 16(3), 241–257.
        15. Allen, R.S.; Hanif, A.M.; Gogniat, M.A. et al. TrkB signalling pathway mediates the protective effects of exercise in the diabetic rat retina. Eur J Neurosci. 2018, 47(10), 1254–1265.
        16. Cui, J.Z.; Wong, M.; Wang, A. et al. Exercise inhibits progression of diabetic retinopathy by reducing inflammatory, oxidative stress, and ER stress gene expression in the retina of db/db mice. Invest Ophthalmol Vis Sci. 2016, 57(12), 5434.
        17. Lu, Y.; Dong, Y.; Tucker, D. et al. Treadmill exercise exerts neuroprotection and regulates microglial polarization and oxidative stress in a streptozotocin-induced rat model of sporadic alzheimer’s disease. J Alzheimers Dis. 2017, 56(4), 1469–1484.
        18. Schneider, S.H.; Khachadurian, A.K.; Amorosa, L.F. et al. Tenyear experience with an exercise-based outpatient life-style modification program in the treatment of diabetes mellitus. Diabetes Care. 1992, 15(11), 1800–1810.
        19. Colberg, S.R. Exercise and diabetes: a clinician’s guide to prescribing physical activity, 1st edn. American Diabetes Association, 2013. Alexandria.
        20. Graham, C.; Lasko-McCarthey, P. Exercise options for persons with diabetic complications. Diabetes Educ. 1990, 16(3), 212–220.
        21. Hamdy, O.; Goodyear, L.J.; Horton, E.S. Diet and exercise in type 2 diabetes mellitus. Endocrinol Metab Clin North Am. 2001, 30(4), 883–907.

        Section 8.

        We have completed the information.

        Research using animal models of diabetes has shown that resistance exercise can cause increased muscle mass [1]. The skeletal muscle is a substantial glucose reservoir in the body and physical exercise is a potent stimulant of glucose uptake partly via the skeletal muscle glucose transporter protein action [2]. That is why resistance training exerting a direct effect on skeletal muscle is likely to play a role in the management of D2T patients [3].

        Resistance training is a good alternative to aerobic exercise. This type of training has not been found to cause a greater number of undesirable events as compared to other types of exercise [3]. Pollock et al. reported that there were more complications during jogging and walking than in strength training in the elderly [4].

        Compliance with recommendations related to aerobic exercise can be difficult. Aerobic training should be conducted almost every day or even all the week to be effective. However, progressive resistance exercise can be effective if performed only three times a week [3].

        Moreover, some coexisting diseases in diabetic patients (cardiovascular and peripheral vascular disorders, neuropathy and motor impairment – foot ulceration, claudication and a risk of falling down) may hinder aerobic performance. In all these situations, resistance training is not only a real alternative but can also be more viable than aerobic exercise. In patients who can safely take aerobic exercise, a combined training program is more effective than the progressive resistance training program alone [3]. Sigal et al. reported that the levels of glycosylated hemoglobin decreased markedly in the combined training group as compared to the only aerobic or resistance training groups [5].

        Eight weeks with two or three 45 min. sessions of progressive training are sufficient to improve glycemic control [3].

        A reduction in glycosylated hemoglobin by 1% is associated with a 37% decrease in the risk of microvascular complications and a 21% decrease in the risk of diabetes-related death [6].

        Snowling and Hopkins observed a 0.5% reduction in glycosylated hemoglobin after progressive resistance training [7].

        1. Farrell, P.; Fedele, M.; Hernandez, J.; Fluckey, J.; Miller, J.; Lang, C. et al. Hypertrophy of skeletal muscle in diabetic rats in response to chronic resistance exercise. Journal of Applied Physiology. 1999, 87, 1075–1082.
        2. Schuller, G.; Linke, A. Diabetes, exercise. In Goldstein, B.; Muller-Wieland, D. (Eds) Type 2 Diabetes: Principles, Practice. New York: Informa Healthcare, Ch 6. 2008
        3. Irvine, C.; Taylor, N.C. Progressive resistance exercise improves glycaemic control in people with type 2 diabetes mellitus: a systematic review. Australian Journal of Physiotherapy. 2009, 55, 237-246.
        4. Pollock, M.; Carroll, J.; Graves, J.; Leggett, S.; Braith, R.; Limacher, M. et al. Injuries, adherence to walk/jog, resistance training programs in the elderly. Medicine, Science in Sports, Exercise. 1991, 23, 1194–1200.
        5. Sigal, R.; Kenny, G.; Boule, N.; Wells, G.; Prud’homme, D.; Fortier, M. et al. Effects of aerobic training, resistance training, or both on glycemic control in type 2 diabetes: A randomized trial. Annals of Internal Medicine. 2007, 147, 357–369.
        6. Stratton, I.; Adler, A.; Neil, H.; Matthews, D.; Manley, S.; Cull, C. et al. Association of glycemia with macrovascular, microvascular complications of type 2 diabetes (UKPDS35): Prospective observational study. BMJ. 2000, 321, 405–412.
        7. Snowling, N.; Hopkins, W. Effects of different modes of exercise training on glucose control, risk factors for complications in type 2 diabetic patients: a meta-analysis. Diabetes Care. 2006, 29, 2518–2527.

        Section 11.

        Dodaliśmy zalecane informacje.

        Chlorogenic acid (CGA) is a polyphenol found in a variety of products, such as coffee, grains, potatoes and apples. It is the ester of caffeic and quinic acids, showing antibacterial, antiinflammatory, antioxidant and anineoplastic effects [1-3]. CGA can delay glucose absorption in the small intestine and decrease glucose production in the liver [4,5]. It is believed that CGA stimulates secretion of glucagon-like peptide 1 known to exert a beneficial effect on the response to glucose in pancreatic beta cells [3]. In the liver, CGA inhibits glucose-6-phospatase, thus decreasing hepatic glucose production [5,6]. CGA reduces hyperpermeability of retinal vessels in diabetic rats. The diabetic rats with higher level of VEGF and down-regulation of occludin, claudin-5 and ZO-1 showed breakdown of the blood-retina barrier and intensified vascular leakage. CGA managed to maintain occludin expression and reduced the levels of VEGF, which decreased the BRB breakdown and inhibited vascular leakage. Thus, CGA may prevent BRB breakdown in retinopathy [7].

        1. Dos Santos, M.D.; Almeida, M.C.; Lopes, N.P.; de Souza, G.E. Evaluation of the anti-inflammatory, analgesic and antipyretic activities of the natural polyphenol chlorogenic acid. Biol Pharm Bull. 2006, 29, 2236-2240.
        2. Puupponen-Pimiä, R.; Nohynek, L.; Meier, C.; Kähkönen, M.; Heinonen, M.; Hopia, A, Oksman-Caldentey, K.M. Antimicrobial properties of phenolic compounds from berries. J Appl Microbiol. 2001, 90, 494-507.
        3. Ma, C.M.; Kully, M.; Khan, J.K.; Hattori, M.; Daneshtalab, M. Synthesis of chlorogenic acid derivatives with promising antifungal activity. Bioorg Med Chem. 2007, 15, 6830-6833.
        4. McCarty, M.F. A chlorogenic acid-induced increase in GLP-1 production may mediate the impact of heavy coffee consumption on diabetes risk. Med Hypotheses. 2005, 64, 848-853.
        5. Arion, W.J.; Canfield, W.K.; Ramos, F.C. Schindler, P.W.; Burger, H.J.; Hemmerle, H.; Schubert, G.; Below, P.; Herling, AW. Chlorogenic acid and hydroxynitrobenzaldehyde: new inhibitors of hepatic glucose 6-phosphatase. Arch Biochem Biophys. 1997, 339, 315-322.
        6. Herling, A.W.; Burger, H.; Schubert, G.; Hemmerle, H.; Schaefer, H.; Kramer, W. Alterations of carbohydrate and lipid intermediary metabolism during inhibition of glucose-6-phosphatase in rats. Eur J Pharmacol. 1999, 386, 75-82.
        7. Shin, J.Y.; Sohn, J.; Hyung, K. ParkChlorogenic Acid Decreases Retinal Vascular Hyperpermeability in Diabetic Rat Model. Korean Med Sci .2013, 28, 608-613.

        Calcium dobesilate (CaD) is considered to be an angioprotective drug. For many years CaD was used in diabetic retinopathy [4,5]. Researchers investigating its efficacy are divided. A few clinical studies showed delayed progression of diabetic retinopathy after long-term oral treatment with CaD. It was found to inhibit changes in tight junction proteins, ICAM-1 and leukocyte adhesion to retinal vessels which are known to lie at the base of growing permeability of the blood-retina barrier. These results were correlated with the inhibition of oxidative / nitrosative stress and the activity of  p38 MAPK and NF-κB [1,2]. A positive effect of CaD (2000 mg/day for 2 years) was found in patients suffering from early diabetic retinopathy [6]. Other studies failed to show a positive effect of CaD on the retina of diabetic patients [8,9].  In a recent study (CALDIRET) (30), in which patients with mild to moderate non-proliferative diabetic retinopathy were followed-up for 5 years, CaD did not reduce diabetic macular edema [7].

        1. Leal, E.C.; Martins, J.; Voabil, P.; Liberal, J.; Chiavaroli, C.; Bauer, J.; Cunha-Vaz, J.; Ambrósio, A.F. Calcium Dobesilate Inhibits the Alterations in Tight Junction Proteins and Leukocyte Adhesion to Retinal Endothelial Cells Induced by Diabetes 2010, 59(10), 2637–2645. Published online 2010 Jul 13. doi: 10.2337/db09-1421
        2. Adamis, A.P.; Berman, A.J. Immunological mechanisms in the pathogenesis of diabetic retinopathy. Semin Immunopathol. 2008, 30, 65–84. [PubMed] [Google Scholar]
        3. Kern, T.S. Contributions of inflammatory processes to the development of the early stages of diabetic retinopathy. Exp Diabetes Res. 2007, 2007, 95103. [PMC free article] [PubMed] [Google Scholar]
        4. Berthet, P.; Farine, J.C.; Barras, J.P. Calcium dobesilate: pharmacological profile related to its use in diabetic retinopathy. Int J Clin Pract. 1999, 53, 631–636. [PubMed] [Google Scholar]
        5. Tejerina, T.; Ruiz, E. Calcium dobesilate: pharmacology and future approaches. Gen Pharmacol. 1998, 31, 357–360.
        6. Ribeiro, M.L.; Seres, A.I.; Carneiro, A.M.; Stur, M.; Zourdani, A.; Caillon, P.; Cunha-Vaz, J.G. DX-Retinopathy Study Group Effect of calcium dobesilate on progression of early diabetic retinopathy: a randomised double-blind study. Graefes Arch Clin Exp Ophthalmol. 2006, 244, 1591–1600.
        7. Haritoglou, C.; Gerss, J.; Sauerland, C.; Kampik, A.; Ulbig, M.W. CALDIRET study group Effect of calcium dobesilate on occurrence of diabetic macular oedema (CALDIRET study): randomised, double-blind, placebo-controlled, multicentre trial. Lancet. 2009, 373, 1364–1371.
        8. Liu, J.; Li, S.; Sun, D. Calcium Dobesilate and Micro-vascular diseases. Life Sci. 2019, Mar 15, 221, 348-353. doi: 10.1016/j.lfs.2019.02.023. Epub 2019 Feb 12.
        9. Solà-Adell, C.; Bogdanov, P.Hernández, C.; Sampedro, J.; Valeri, M.; Garcia-Ramirez, M.; Pasquali, C.; Simó, R. Calcium Dobesilate Prevents Neurodegeneration and Vascular Leakage in Experimental Diabetes. Curr Eye Res. 2017, 42(9), 1273-1286. doi: 10.1080/02713683.2017.1302591. Epub 2017 Jun 2

        It has been found that n3-PUFA can reduce the number of retinal acellular capillaries in diabetes and inflammatory markers in the retina of diabetic animals [1,2]. Diabetic patients taking n-3 PUFA supplements for 18 months had lower risk of retinopathy [3].

        The effect of vitamin C intake on DR is not certain [4-6]. The use of vitamin C supplements may suggest that this vitamin is involved in retinopathy prevention [4].

        Millen et al. [5], conducting a prospective study of the ARIC participants,  failed to observe an effect of vitamin C and E intake from food alone or a combination of food and supplements on the risk of retinopathy. However, they found an interaction between race and vitamin E; high consumption of vitamin E from food alone, or food combined with supplements, was associated with higher prevalence of retinopathy in Caucasians, but not in African Americans [5]. Diminished likelihood of retinopathy was noted in those taking (C 3 years) vitamin C, E, and multivitamin supplements in comparison with non-users [5]. However, in a cross-sectional analysis of NHANES III participants, Millen et al. found no relationship between taking vitamin C or E supplements for a long time (C 5 vs.\1 year) and the presence of retinopathy [7].

        The intake of vitamin D and fish oil supplements was not associated with the risk of retinopathy in the ARIC [8].

        Lutein intake was not significantly correlated with retinopathy [9]. In one study, the serum level of lutein was found to be decreased in patients with nonproliferative diabetic retinopathy as compared to diabetics without retinopathy [10].

        Another study suggested a positive impact of high plasma level of lutein/zeaxanthin and lycopene on the risk of diabetic retinopathy [11]. However, increased consumption of b-carotene was found to be related to lower risk and was not associated with retinopathy [4,6].

        1. Tikhonenko, M.; Lydic, T.A.; Opreanu, M.; Li, C.S.; Bozack, S.; McSorley, K.M. et al. N-3 polyunsaturated Fatty acids prevent diabetic retinopathy by inhibition of retinal vascular damage and enhanced endothelial progenitor cell reparative function. PLoS ONE. 2013, 8(1), e55177.
        2. Shen, J.H.; Ma, Q.; Shen, S.R.; Xu, G.T.; Das, U.N. Effect of alphalinolenic acid on streptozotocin-induced diabetic retinopathy indices in vivo. Arch Med Res. 2013, 44(7), 514–520.
        3. Roig-Revert, M.J.; Lleo-Perez, A.; Zanon-Moreno, V.; Vivar-Llopis, B.; Marin-Montiel, J.; Dolz-Marco, R. et al. Enhanced oxidative stress and other potential biomarkers for retinopathy in type 2 diabetics: beneficial effects of the nutraceutic supplements. Biomed Res Int. 2015, 2015, 408180.
        4. Tanaka, S.; Yoshimura, Y.; Kawasaki, R.; Kamada, C.; Tanaka, S.; Horikawa, C. et al. Fruit intake and incident diabetic retinopathy with type 2 diabetes. 2013, 24(2), 204–201.
        5. Millen, A.E.; Klein, R.; Folsom, A.R.; Stevens, J.; Palta, M.; Mares, J.A. Relation between intake of vitamins C and E and risk of diabetic retinopathy in the Atherosclerosis Risk in Communities Study. Am J Clin Nutr. 2004, 79(5), 865–873.
        6. Mayer-Davis, E.J.; Bell, R.A.; Reboussin, B.A.; Rushing, J.; Marshall, J.A.; Hamman, R.F. Antioxidant nutrient intake and diabetic retinopathy: the San Luis Valley Diabetes Study. Ophthalmology. 1998, 105(12), 2264–2270.
        7. Millen, A.E.; Gruber, M.; Klein, R.; Klein, B.E.; Palta, M.; Mares, J.A. Relations of serum ascorbic acid and alpha-tocopherol to diabetic retinopathy in the Third National Health and Nutrition Examination Survey. Am J Epidemiol. 2003, 158(3), 225–233.
        8. Millen, A.E.; Sahli, M.W.; Nie, J.; LaMonte, M.J.; Lutsey, P.L.; Klein, B.E. et al. Adequate vitamin D status is associated with the reduced odds of prevalent diabetic retinopathy in African Americans and Caucasians. Cardiovasc Diabetol. 2016, 15(1), 128.
        9. Sahli, M.W.; Mares, J.A.; Meyers, K.J.; Klein, R.; Brady, W.E.; Klein, B.E. et al. Dietary intake of lutein and diabetic retinopathy in the atherosclerosis risk in Communities Study (ARIC). Ophthalmic Epidemiol. 2016, 23(2), 99–108.
        10. Koushan, K.; Rusovici, R. Li, W.; Ferguson, L.R.; Chalam, K.V. The role of lutein in eye-related disease. 2013, 5(5), 1823–1839.
        11. Brazionis, L.; Rowley, K.; Itsiopoulos, C.; O’Dea, K. Plasma carotenoids and diabetic retinopathy. Br J Nutr. 2009, 101(2), 270–277.